# Spatial transcriptomics reveal markers of histopathological changes in Duchenne muscular dystrophy mouse models

L.G.M. Heezen [1], T. Abdelaal [2,3,4], M. van Putten[1], A. Aartsma-Rus [1], A. Mahfouz [1,4,5] & P. Spitali [1]

Duchenne muscular dystrophy is caused by mutations in the *DMD* gene, leading to lack of dystrophin. Chronic muscle damage eventually leads to histological alterations in skeletal muscles. The identification of genes and cell types driving tissue remodeling is a key step to developing effective therapies. Here we use spatial transcriptomics in two Duchenne muscular dystrophy mouse models differing in disease severity to identify gene expression signatures underlying skeletal muscle pathology and to directly link gene expression to muscle histology. We perform deconvolution analysis to identify cell types contributing to histological alterations. We show increased expression of specific genes in areas of muscle regeneration (*Myl4*, *Sparc*, *Hspg2*), fibrosis (*Vim*, *Fn1*, *Thbs4*) and calcification (*Bgn*, *Ctsk*, *Spp1*). These findings are confirmed by smFISH. Finally, we use differentiation dynamic analysis in the D2-*mdx* muscle to identify muscle fibers in the present state that are predicted to become affected in the future state.

Duchenne muscular dystrophy (DMD) is an X-linked recessive and fatal disorder affecting approximately one in 5000 male births[1]. Out-of-frame mutations in the *DMD* gene on chromosome Xp21 lead to the absence of dystrophin protein[2,3]. Dystrophin is part of the dystrophin-associated glycoprotein complex and acts as a linker between the extracellular matrix and the cytoskeleton, aiding in muscle membrane stability. The absence of dystrophin leads to a weakened sarcolemma, rendering the muscle fibers more vulnerable to contraction-induced damage[4–7].

Muscle fibers from early-stage DMD patients undergo cycles of degeneration and compensatory regeneration. Later, once the regenerative capacity of the muscle fibers becomes exhausted, they are substituted by fibrotic and adipose tissue[8–10]. Due to the accumulation of fibrotic and adipose tissue and subsequent progressive loss of muscle function[11], DMD patients become wheelchair dependent by 11–12 years of age and decease in their third or fourth decade due to cardiorespiratory failure[12,13].

Histological and molecular analyses of muscle samples obtained from DMD patients and animal models enabled the description of alterations resulting from a lack of dystrophin. Studies in humans showed how necrosis and inflammation promote the accumulation of fibrosis and fat infiltration[11,14,15]. These observations were supported by bulk proteomics[16] and transcriptomics studies[17]. Preclinical studies in the *mdx* mouse model (carrying a nonsense mutation in exon 23 of the *Dmd* gene) and especially in the more severely affected D2-*mdx* mouse (same mutation on a DBA/2J background), revealed the presence of necrosis, extensive inflammation, regeneration, central nucleation, changes in fiber size, fibrosis and calcified tissue[18–20]. Various studies have assessed gene expression changes related to the disease. An upregulation of genes involved in inflammation (*Lgals3*, *Cd68*), muscle regeneration (*Myog*, *Pax7*, *Myh3*) and fibrosis (*Col1a1*, *Lox*) has been identified in preclinical models[15,21–24]. These studies pointed out how specific cell types, such as fibro-adipogenic progenitors (FAPs) and profibrotic factors (e.g. TGF-ß, CTGF and SPP1) play an important role

[1]Department of Human Genetics, Leiden University Medical Center, Leiden, The Netherlands. [2]Department of Radiology, Leiden University Medical Center, Leiden, The Netherlands. [3]Systems and Biomedical Engineering Department, Faculty of Engineering Cairo University, Giza, Egypt. [4]Delft Bioinformatics Lab, Delft University of Technology, Delft, The Netherlands. [5]Leiden Computational Biology Center, Leiden University Medical Center, Leiden, The Netherlands. ✉e-mail: p.spitali@lumc.nl

in regeneration and are thought to be main contributors of intra-muscular fibrotic infiltration in dystrophic muscles[25–30].

Overall, associations between histological alterations with cell types and gene expression have been suggested. However, there are no studies where histological, cellular and gene expression data have been simultaneously and spatially investigated on the same tissue section. This limitation can be overcome by using spatial transcriptomics (ST), which allows to directly link histology to gene expression. Here, we applied ST to identify molecular markers that underlie the histopathological changes observed in *mdx* and D2-*mdx* mice. We identified clusters based on gene expression profiling and histological features. Using spot deconvolution analysis, we show how different cell types contribute to the gene expression signature in tissue sections with 55 μm resolution. Moreover, we identified differentially expressed marker genes that are involved in histopathological changes (regeneration, fibrosis and calcification). These findings were confirmed with a more sensitive approach that holds a higher spatial resolution, smFISH (RNAscope). Finally, we predicted differentiation dynamics within the muscle section of the D2-*mdx* muscle using RNA velocity to assess whether we could identify a pattern predicting the future state of muscle fibers in tissue that is heavily affected by tissue remodeling[31]. Our study directly maps molecular changes to tissue alterations in preclinical models of DMD at spatial resolution. The mapping of these data deepens our knowledge of DMD pathology.

## Results

### Spatial transcriptomics reveals clusters related to histology

Quadriceps muscle of healthy (C57BL10 and DBA/2J) and dystrophic (*mdx* and D2-*mdx*) mice were analyzed. The four muscles composing the quadriceps group (rectus femoris (RF), vastus lateralis (VL), vastus medialis (VM) and vastus intermedius (VI)) were present on the slide, although the VI was not present in all sections. Muscles of healthy mice showed homogenous tissue composition, with similarly sized muscle fibers, shown by Hematoxylin and Eosin (HE) staining compared to dystrophic tissues, where variation in fiber size, necrosis, inflammation, fibrosis, regeneration (*mdx*) and calcification (D2-*mdx*) were observed (Fig. 1). After processing, a total of 474,186,322 paired-end reads covering 7028 spots were obtained with a median of 1122 genes per spot (Table 1).

Spots were clustered for each sample independently. Based on gene expression profiles, marker gene expression and histological observations, clusters were annotated (see Methods for details). Clusters representing healthy muscle fibers were identified in the wildtype models. Top markers for healthy muscle fibers included muscle specific genes, such as *Myh4*, *Mylpf*, and *Ckm*. We also identified clusters representing the perimysium and other connective tissue, expressing extracellular matrix genes, such as *Col1a2*, *Col1a1*, and *Fmod*. A VI specific cluster was defined by the expression of myosins specific for slow twitch muscle fibers (*Myl2*, *Myl3*, *Myh7*) (Fig. 1a, Supplementary Fig. 1). Additionally, we identified smaller clusters enriched in genes specific for erythrocytes (*Hba-bt*, *Hbb-a2*, *Hba-a1*) and immune response (*Lyz2*, *Spp1*, *Ctsb*). The latter one was observed in an area of damaged tissue in the C57BL10 mouse only, underpinning the strength of using spatial transcriptomics to identify small tissue alterations (Supplementary Fig. 1). An additional small cluster enriched for neuromuscular junction (NMJ) related genes (*Mpz*, *Mbp*, *Chrne*) was found in the DBA/2J muscle (Fig. 1b, Supplementary Fig. 2). Some clusters (muscle fibers, erythrocytes, connective tissue) were present in all mouse models with comparable expression levels of top marker genes (Supplementary data files 1 and 2).

Dystrophic samples displayed additional clusters, not present in the control samples, matching the histopathological changes present in these samples. *Mdx* mice showed mature muscle fibers (*Amd1*, *Amd2*, *Smox*), regenerated fibers (*Gsn*, *Fhl1*, *Igfbp7*), regenerating fibers with inflamed patches (*Xirp2*, *Lars2*, *Mybpc1*) and some necrotic muscle fibers with infiltrating macrophages (*Cd68*, *Vim*, *Tyrobp*) (Fig. 1c). Additionally, a small cluster of adipocytes was identified (*Apoc1*, *Scd1*, *Adipoq*) (Supplementary Fig. 3). The more severely affected D2-*mdx* mouse additionally showed a large cluster with prominent inflammation and calcification signatures (*Spp1*, *Mgp*, *Mpeg1*) as well as necrotic fibers (*UbaS2*, *Eef1d*, *Cox6a2*) (Fig. 1d, Supplementary Fig. 4).

### Validation of the NMJ cluster

To verify the NMJ cluster in the DBA/2J sample, staining with α-bungarotoxin (BTX) was performed on a consecutive section of the DBA/2J muscle (Fig. 2a). BTX binds to the nicotinic acetylcholine receptor found at the NMJ. The BTX signal was isolated and overlayed on the HE-stained image of the DBA/2J sample to assess localization of the NMJs detected by staining (Fig. 2c). Comparison of the NMJ annotated spots with the BTX signal showed overlap (Fig. 2b) and validated the detection of the NMJ cluster at the location of these spots.

The absence of a specific NMJ cluster in the other mouse models (C57BL10, *mdx* and D2-*mdx*) was explored by considering the expression of the NMJ marker genes in all mouse models (Supplementary Fig. 5). The C57BL10 sample shows that NMJ marker genes are present at specific locations, suggesting the presence of NMJs. However, the expression of NMJ marker genes is fairly low and scattered throughout the *mdx* and D2-*mdx* samples. This may hint to a dysregulated NMJ gene expression signature in dystrophic mice. The NMJ alteration in these mice was assessed by BTX staining, confirming altered structure or absence of NMJs on consecutive sections obtained in these mice (Supplementary Fig. 6). BTX staining on the D2-*mdx* and *mdx* samples shows NMJs with altered and fragmented structure, which has previously been described as a characteristic of Duchenne mouse models[32]. The BTX staining on a consecutive section of the C57BL10 sample reveals that the NMJ location matches the location of the plotted NMJ marker genes, similarly to the DBA/2J sample.

### Unravelling the spot: enrichment of cell types in annotated clusters

The resolution of the Visium slides does not allow profiling gene expression at the single cell level. To overcome this and to estimate the contribution of different cell types to the observed transcriptomic signatures, we deconvoluted the spots using a snRNAseq reference dataset. This dataset was obtained from the tibialis anterior muscle of wildtype mice and a mouse model carrying a deletion of *Dmd* exon 51 (ΔEx51) resulting in dystrophin absence and histological alterations similar to those observed in *mdx* mice[33]. The selected reference dataset includes nuclei originating from muscle as well as non-muscle nuclei. These nuclei were annotated, allowing for deconvolution of several cell types (Fig. 3a).

Deconvolution showed how the different cell types contribute consistently to the gene expression signature. The main contributing cell types were type IIx myonuclei (ranging from 46.6% to 59.9% of nuclei across the 4 quadriceps muscles, Fig. 3b), followed by type IIb myonuclei (ranging from 17.6% to 24.3% of nuclei). A consistent increase in macrophages (MPH) was observed in *mdx* and D2-*mdx* mice compared to background matched wildtype mice. Furthermore, the muscle regenerative capacity characteristic of the BL10 background was reflected by elevated levels of myoblasts, muscle satellite cells and regenerative myonuclei compared to the mice on the DBA/2J background known to have reduced regeneration potential.

The average cell type contribution to spots assigned to each specific cluster was calculated to assess the enrichment for specific cell types across annotated clusters. Figure 3c shows how the connective tissue cluster in the C57BL10 muscle matches the distribution of markers (*Col1a2*, *Col1a1* and *Fmod*) associated with the connective tissue, tenocytes (TC) and the myotendinous junction myonuclei (MTJ). Deconvolution showed an enrichment for these cell types at

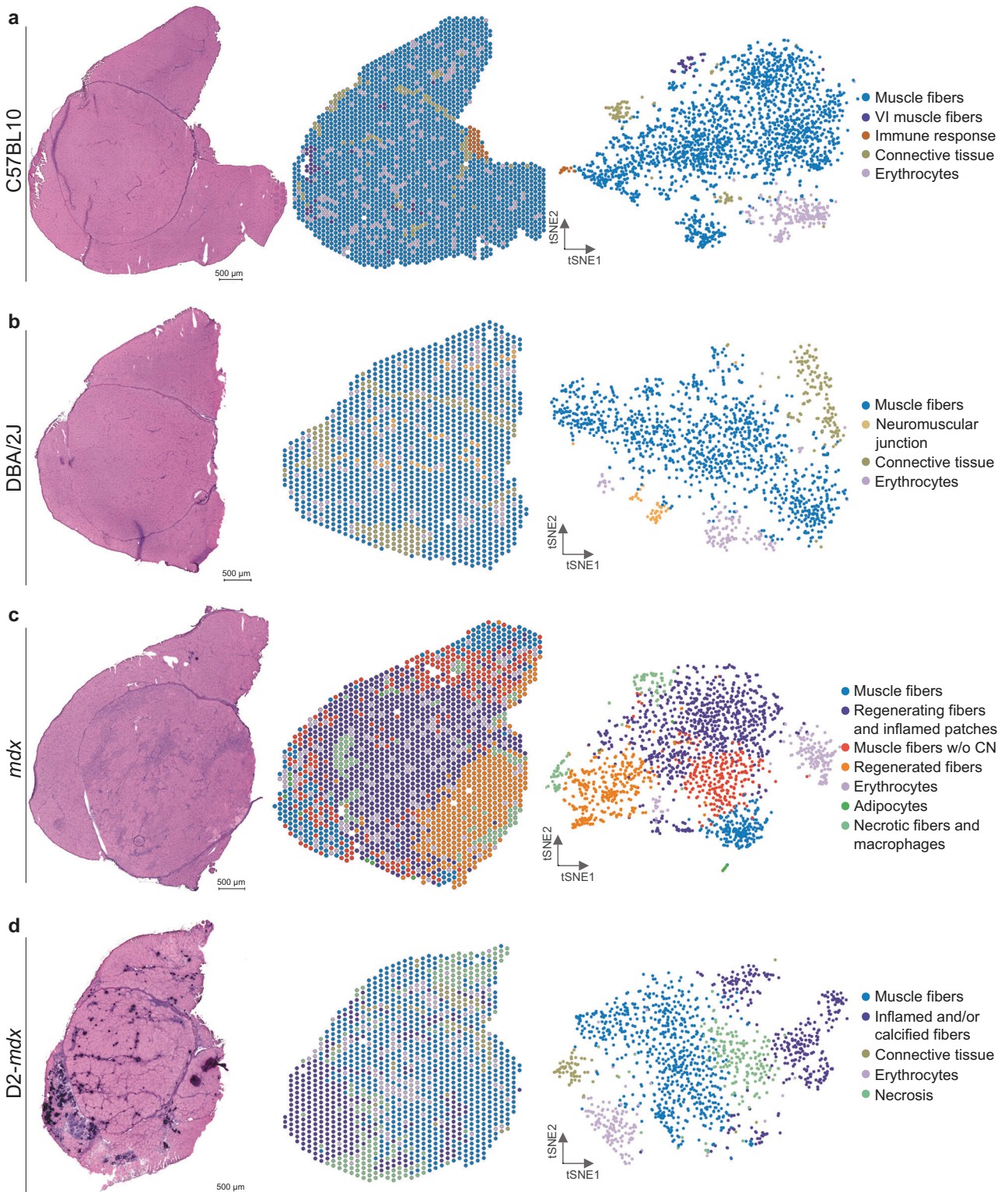

**Fig. 1 | Characterizing muscle tissue in DMD and wildtype mouse models using spatial transcriptomics.** (left) the HE stained section, (middle) displays the Visium spots spatially plotted and colored by cluster, and (right) a tSNE map of the Visium spots colored by cluster for (**a**) C57BL10, (**b**) DBA/2J, (**c**) *mdx* and (**d**) D2-*mdx*.

these locations, with 12.9% of nuclei being TC and 8.5% of nuclei being MTJ compared to an overall respective average of 1.4% and 2.2% per spot in the C57BL10 sample.

Another example is given by zooming into the neuromuscular junction (NMJ) cluster in the DBA/2J tissue section (Fig. 3d).

Mapping the specific marker genes for the NMJ myonuclei (*Mpz, Pmp22* and *Chrne*) shows increased expression in the cluster spots. This is also reflected upon deconvolution, where the NMJ myonuclei are a clear contributing factor for the spots included in these clusters. The average percentage of NMJ myonuclei per spot

was 0.7% for the DBA/2J model, while being 17.8% in these selected spots.

Next, we focused on the clusters that showed clear histopathological changes in both the *mdx* and D2-*mdx* mice (necrotic fibers and macrophages, inflamed and/or calcified fibers) (Fig. 3e, f). Macrophage markers (*Cd68* for M1-macrophages, *Cd14* for M2-macrophages and *Tgfbi* as a secretion marker of M2-macrophages) were plotted accompanied by necrosis specific markers (*Tnfrsf1a* and *Tnfrsf1b*). The expression pattern of these markers spatially aligned with the spots in these clusters. In *mdx* mice, an enrichment for MPH in these spots (23.8% of nuclei) was observed compared to the average per spot (5.1%). Finally, in D2-*mdx* the overall average per spot of MPH contribution was 6.8% and in the assigned cluster 22.3%. Altogether, deconvolution analysis reveals enrichment of specific cell types underlying the observed gene expression patterns at spatial resolution (Supplementary Fig. 7).

### Table 1 | Sequencing results of Visium experiment

| Sample | Number of reads | Number of spots under tissue | Mean reads per spot | Median UMI counts per spot | Median genes per spot |
|---|---|---|---|---|---|
| C57BL10 | 109,031,572 | 2348 | 46,436 | 3836 | 1050 |
| DBA/2J | 123,619,541 | 1426 | 86,690 | 8077 | 1772 |
| *mdx* | 119,361,125 | 1884 | 63,355 | 3238 | 1062 |
| D2-*mdx* | 122,174,084 | 1370 | 89,178 | 1933 | 838 |

## Confirming presence of cell types as suggested by deconvolution

To confirm the enrichment of specific cell types that were suggested by deconvolution analysis, single molecule fluorescent in situ hybridization (smFISH, RNAscope) of cell specific marker genes was performed for a subset of cell types: *Cd34*, *Pdgfra* and *Cd68* were analyzed representing regenerating myonuclei (RegMyon), fibroadipogenic cells (FAPs) and macrophages (MPH), respectively. The presence, or enrichment, of FAPs in affected areas in the D2-*mdx* model (Fig. 4b) is reflected by expression of *Pdgfra* (Fig. 4c). The presence (Fig. 4a-•) and absence (Fig. 4a-×) of FAPs that was suggested by spot deconvolution, was confirmed using this sensitive approach with high spatial resolution (Fig. 4d). Similar analysis was done for the presence of RegMyon and MPHs as depicted in Supplementary Figs. 8 and 9, both confirming the results derived from the applied spot deconvolution strategy.

## Identification of muscle regeneration biomarkers

From 3 to 12 weeks of age, *mdx* muscles undergo repetitive cycles of degeneration and regeneration[34]. To identify genes associated with muscle regeneration we compared areas characterized by the presence of centralized nuclei (CN) with areas with nuclei located at the periphery in the *mdx* mouse (Fig. 5a). Spot selection was based on previously defined clusters (regenerating: regenerated fibers and regenerating fibers and inflamed patches, nonregenerating: muscle fibers without CN). An additional criterion for selection was the expression of *Myog*, *Igfbp7* and *Myh3* in areas of regeneration and absence of these genes in nonregenerating spots (Fig. 5b). Differential gene expression analysis between these two groups of spots revealed

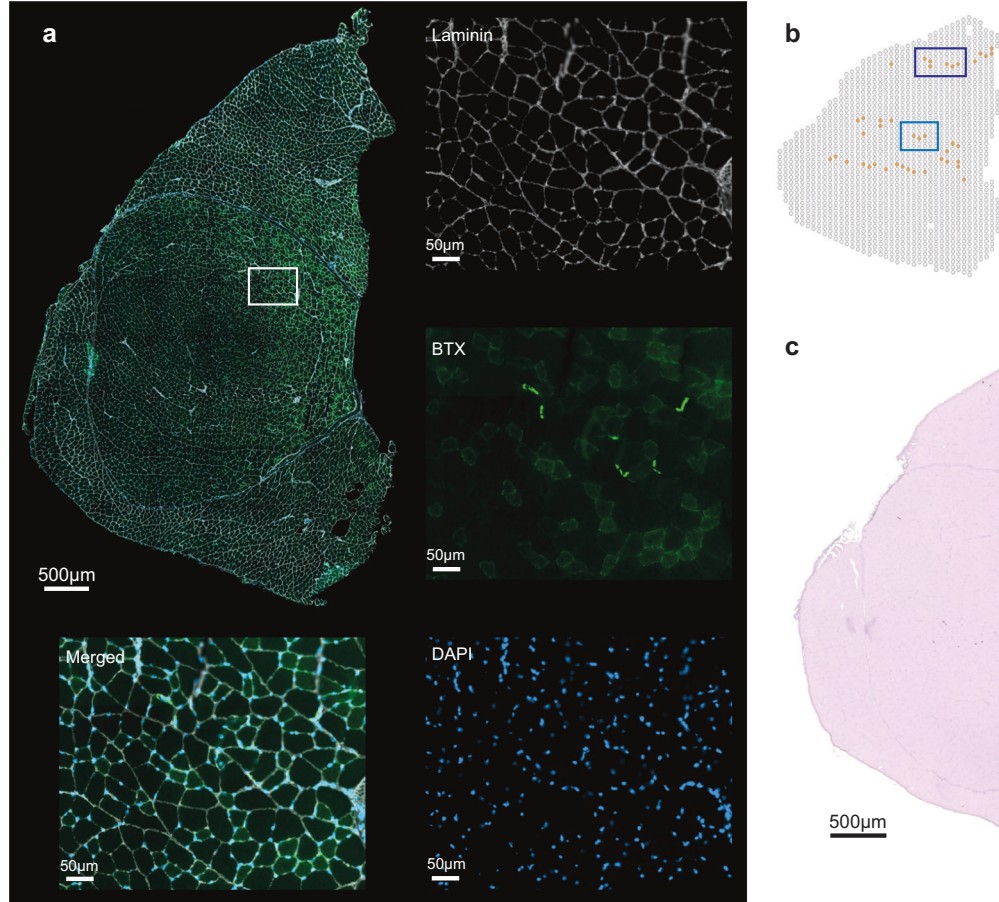

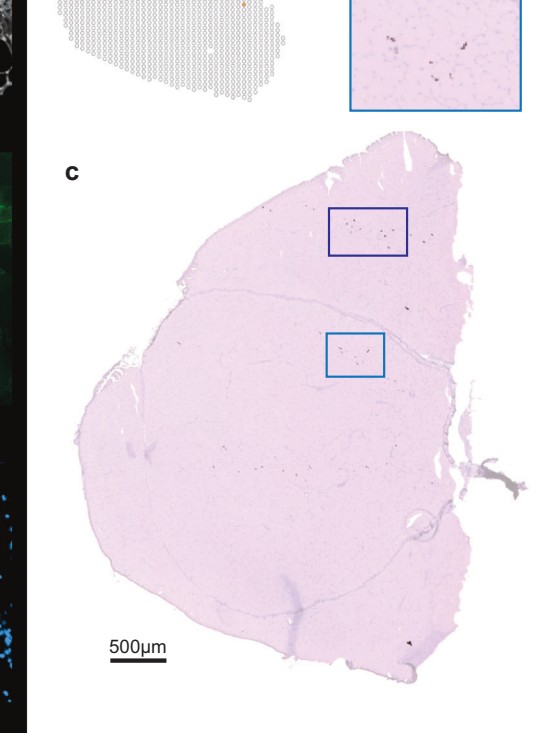

**Fig. 2 | BTX staining in a consecutive section of the DBA/2J Visium sample.** **a** BTX staining on a consecutive DBA/2J section, co-stained for DAPI and laminin with a zoomed-in region displaying several brightly stained NMJs. **b** The Visium spots that were included in the NMJ cluster overlap with the (**c**) BTX isolated signal on the Visium HE stained image (purple and blue highlighted regions).

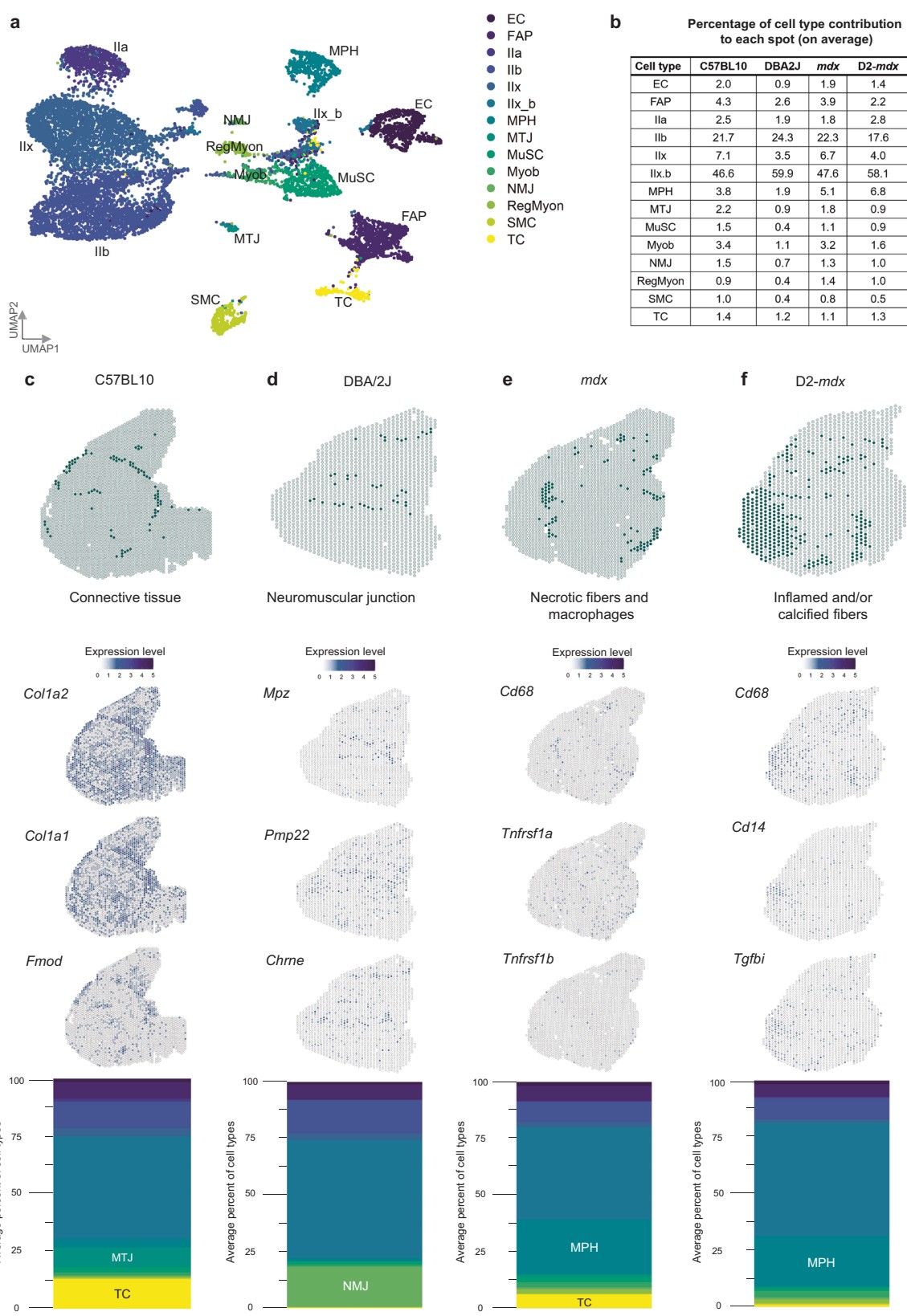

markers of regeneration (all significant differentially expressed markers are summarized in Supplementary data file 3) such as *Myl4*, *Sparc* and *Hspg2*, which showed little-to-no expression in the non-regenerating spots (Fig. 5c). Spatially, the expression of these genes located in areas of regeneration as marked by the histology (Fig. 5d). Besides upregulation of neonatal myosins (*Myl4*, $\log_2$(FC) = 1.42,

adjusted *p* value = 5.50e-17), we also saw upregulation of *Sparc*[35,36] only in areas marked as regenerating muscle fibers ($\log_2$(FC) = 0.90, adjusted *p* value = 5.61e-16, Fig. 5d-*mdx*). Finally, we identify *Hspg2*, a gene involved in cell growth and differentiation, as an important upregulated molecular marker in areas of regeneration ($\log_2$(FC) = 0.57, adjusted *p* value = 1.76e-15). *Hspg2* encodes the

**Fig. 3 | Deconvolution of the spatial data using a snRNAseq reference dataset reveals enrichment of cell types in different mouse models and specific clusters. a** UMAP of the reference dataset including the following cell types: endothelial cells (EC), fibro/adipogenic progenitors (FAP), type IIa myonuclei (IIa), type IIb myonuclei (IIb), type IIx myonuclei (IIx), type IIx_b myonuclei (IIx_b), macrophages (MPH), myotendinous junction myonuclei (MTJ), muscle satellite cells (MuSC), myoblasts (Myob), neuromuscular junction myonuclei (NMJ), regenerative myonuclei (RegMyon), smooth muscle cells (SMC) and tenocytes (TC). **b** The average percentage of cell type contribution per spot in the four mouse models. **c** Deconvolution of the "connective tissue" cluster from C57BL10 showing marker genes *Col1a2*, *Col1a1* and *Fmod* and a stacked barplot displaying the average percentage of contributing cell types to this cluster with an enrichment for MTJ and TC. **d** Deconvolution of the "neuromuscular junction" cluster from DBA/2J plotted marker genes *Mpz*, *Pmp22* and *Chrne* and a stacked barplot displaying the average percentage of contributing cell types to this cluster with an enrichment for NMJ. **e** Deconvolution of the "necrotic fibers and macrophages" cluster from *mdx*, with marker genes *Cd68*, *Tnfrs1a* and *Tnfrs1b* and a stacked barplot displaying the average percentage of contributing cell types to this cluster with an enrichment for MPH and TC. **f** Deconvolution of the "inflamed and/or calcified fibers" cluster from D2-*mdx*, with marker genes *Cd68*, *Cd14* and *Tgfbi* and a stacked barplot displaying the average percentage of contributing cell types to this cluster with an enrichment for MPH.

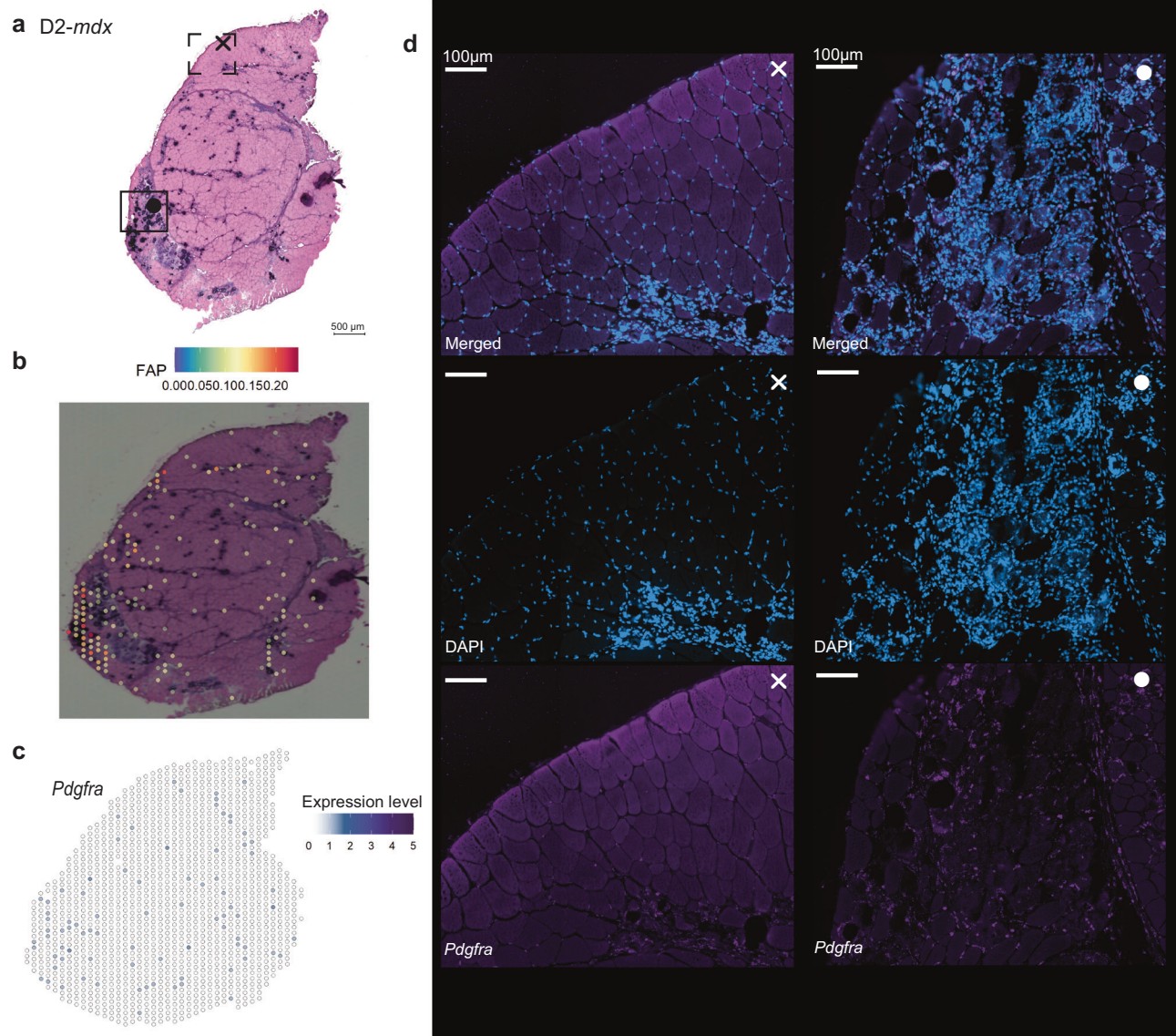

**Fig. 4 | A comparison of FAP cells presence between the deconvolution results and smFISH (RNAscope) validation on the D2-*mdx*. a** D2-*mdx* model with the indications to the approximate location of zoomed-in smFISH images. **b** Location of FAPs based on spot deconvolution. **c** *Pdgfra*, marker gene for FAPs, expression pattern based on Visium data (**d**) smFISH results in two regions that confirm the absence (cross) and presence (filled dot) of FAPs as was expected based on the spot deconvolution results. All scale bars in the immunofluorescent images represent 100 μm.

perlecan protein, which in healthy muscle is present in undamaged sarcolemmal and endomysial areas. However, perlecan expression was found to be increased a few days after muscle damage in regenerating areas, followed by a normalization upon completion of the regeneration process[37,38]. Co-expression of *Hspg2* and *Myl4* was particularly present in areas where active muscle regeneration is taking place in the

mdx model (Supplementary Figs. 12 and 14). To evaluate the specificity of the association of the identified genes with regeneration we assessed the expression of these genes in C57BL10 wildtypes where muscle regeneration was low. While *Myl4* expression was very low in wildtype mice, *Sparc* and *Hspg2* expression was not negligible in wildtype mice. Especially *Sparc* expression was observed at high levels across the

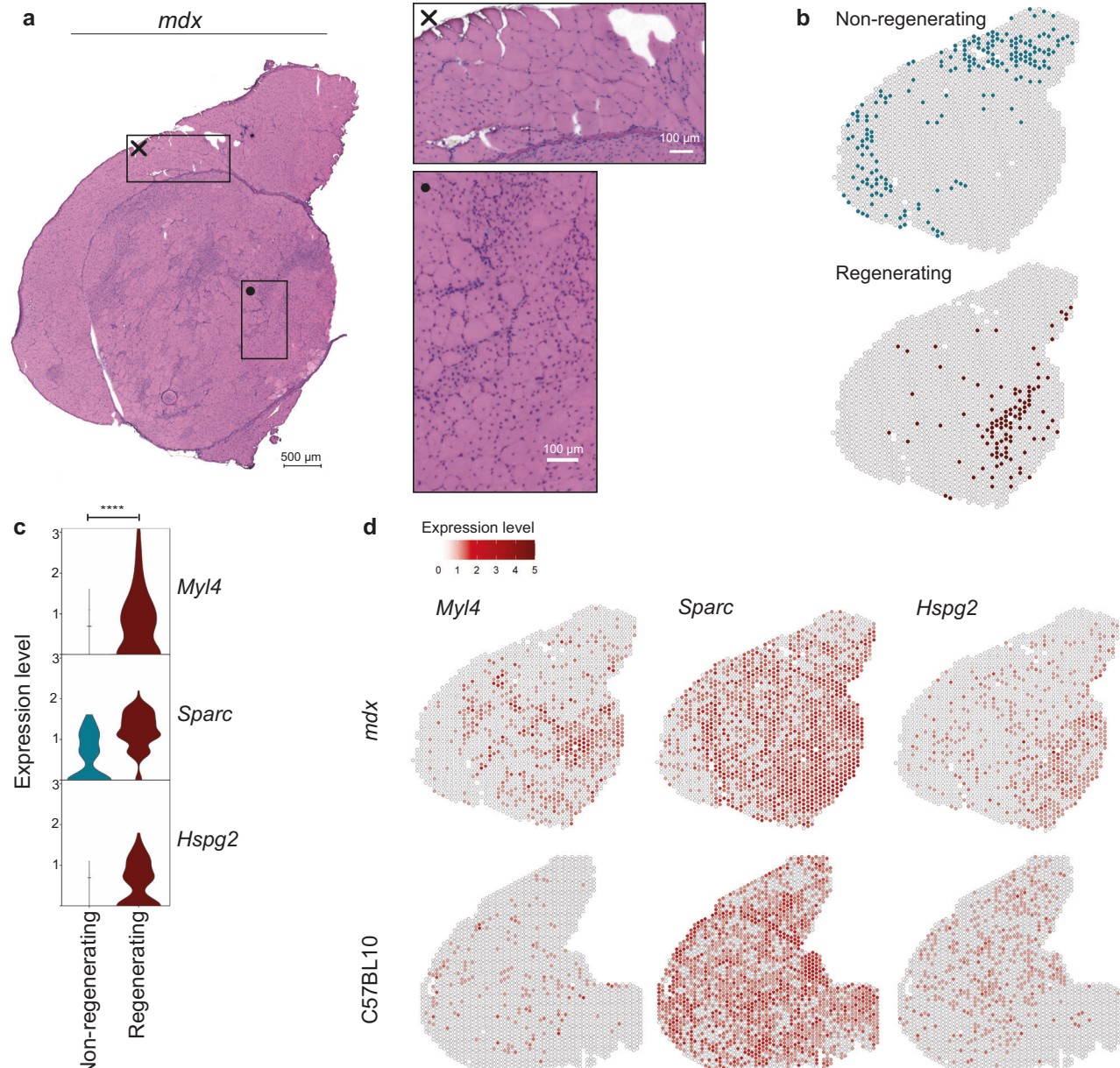

**Fig. 5 | Identifying biomarkers of muscle regeneration in the *mdx* mouse model. a** HE stained *mdx* quadriceps sample with two zoomed-in areas, the cross displaying an area of muscle fibers without centralized nuclei (CN) which are thought to be not regenerated yet, the dot showing an area of recently regenerated or regenerating muscle fibers. **b** Selected spots belonging to the categories "non-regenerating" or "regenerating" were compared for differential gene expression analysis. **c** Expression levels of selected genes *Myl4*, *Sparc* and *Hspg2* in the two categories with an enrichment in the "regenerating" spots in the *mdx* model (**** representing a *p* value < 0.0001). The adjusted *p* values were: 5.5e$^{-17}$ (*Myl4*), 5.6e$^{-16}$ (*Sparc*) and 1.8e$^{-15}$ (*Hspg2*) respectively. **d** Spatially plotted expression of *Myl4*, *Sparc* and *Hspg2* in *mdx* and C57BL10 muscle. Note: expression values should not be compared across models.

tissue, but primarily in the connective tissue sheet and VI muscle (Fig. 5d).

## Identifying genes of severe dystrophic histopathological changes

Given the limited severity of the *mdx* model, we sought to identify genes associated with fibrosis, changes to tissue remodeling and calcification in the D2-*mdx* model. To identify fibrotic markers, spots expressing known fibrotic genes such as *Cd34*, *Lox* and/or *Col1a1* were compared to spots belonging to the muscle fibers cluster where fibrotic genes were not expressed (Fig. 6b). The significant differentially expressed genes are provided as a supplementary file (Supplementary data file 3). Genes such as *Vim*, *Fn1* and *Thbs4* showed significant enrichment in fibrotic regions (Fig. 6d). *Vim* was expressed in non-fibrotic spots, but at significantly lower levels compared to fibrotic spots (log$_2$(FC) = 1.11, adjusted *p* value = 1.71e-10). *Vim* encodes for the protein vimentin expressed in myofibroblasts, and it is expected to be upregulated in areas of skeletal muscle fibrosis[39–41]. Given the role of *Vim* in focal adhesion, where regulatory signals and mechanical force is transmitted between the extracellular matrix (ECM) and an interacting cell[42], expression was found in healthy tissue areas as well in the DBA/2J mouse (Fig. 6e). *Fn1* encoding for fibronectin was identified as biomarker of fibrotic infiltration. Co-expression of *Fn1* and *Vim* is often used to identify fibroblasts and is thus a known marker involved in this histopathological change[43,44]. *Fn1* was highly upregulated in fibrotic spots (log$_2$(FC) = 1.38, adjusted *p* value = 1.80e-16, Fig. 6d) and spatially mapped the locations where tissue damage was most severe (Fig. 6e-D2-*mdx*). Finally, *Thbs4* seemed more specific for fibrosis

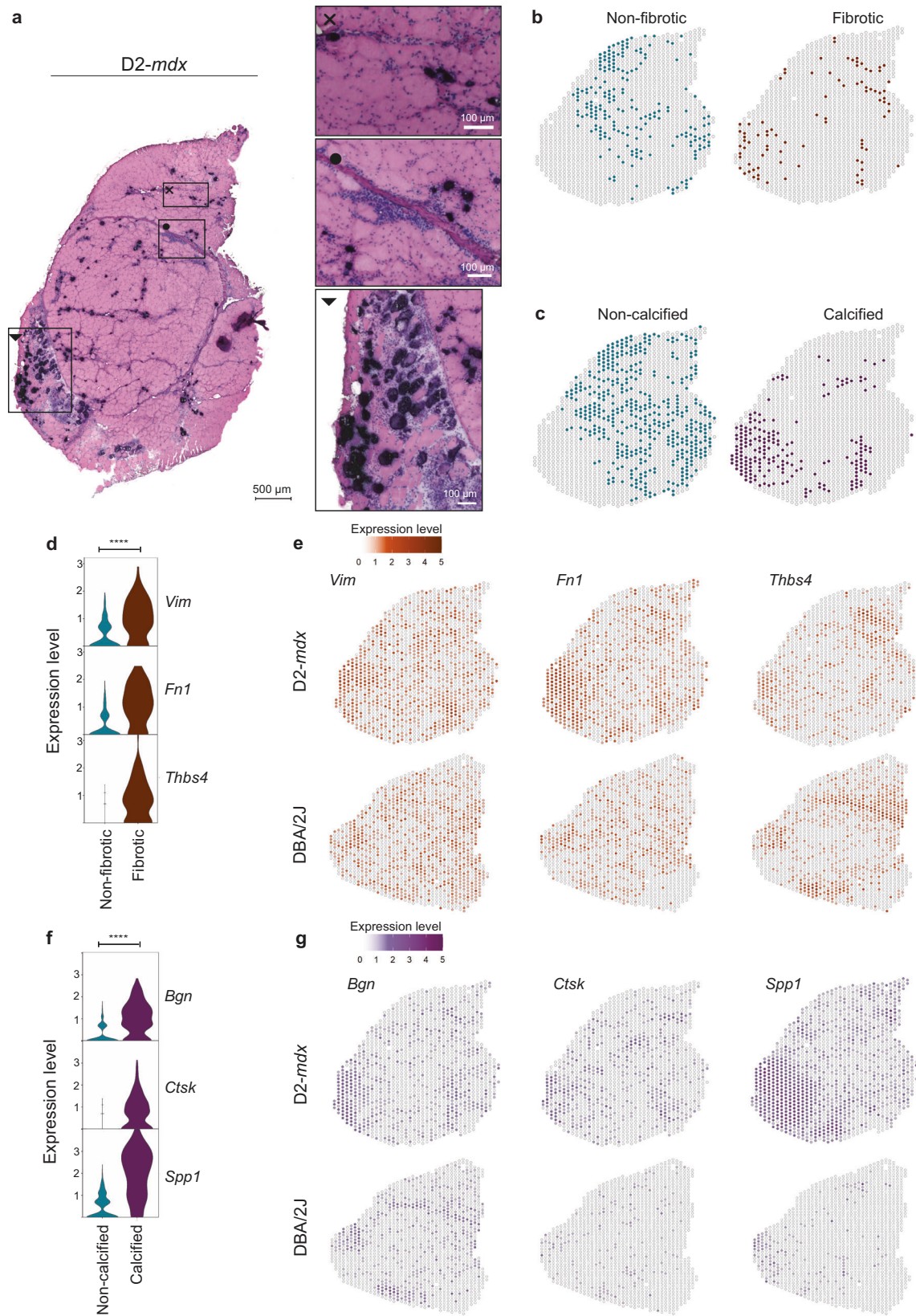

rather than being present in all damaged areas, given the more restricted expression pattern in D2-*mdx* (log$_2$(FC) = 1.32, adjusted p-value = 1.46e-16), whereas localization of *Thbs4* expression is limited to perimysial tissue in the healthy DBA/2J sample (Fig. 6e).

Comparison of calcified and non-calcified spots was performed to identify genes associated with calcification. Spots belonging to the

cluster 'inflamed and/or calcified fibers' and expressing calcification marker *Mgp* were considered calcified, while spots in the muscle fibers cluster lacking *Mgp* expression were considered non-calcified (Fig. 6c). Among the top calcification markers (Supplementary data file 3), we identified *Bgn*, *Ctsk* and *Spp1*. Upregulation of *Bgn* was seen in areas that are highly affected in the D2-*mdx* muscle (log$_2$(FC) = 1.39, adjusted

**Fig. 6 | Differential expression analysis in D2-*mdx* calcified and fibrotic tissue displays upregulated molecular markers. a** HE stained D2-*mdx* QUA sample with three zoomed-in areas, the cross displaying fibrotic infiltration between the myofibers, the dot showing part of the connective tissue sheet with surrounding inflammation, calcification and fibrotic infiltration and the triangle displaying the most severely affected area of the muscle section with extensive calcification, inflammation and necrosis. **b** Selected spots belonging to the categories "non-fibrotic" or "fibrotic" as well as (**c**) "noncalcified" and "calcified" were included in the differential gene expression analysis. **d** Violin plot showing expression levels of selected genes *Vim*, *Fn1* and *Thbs4* which were significantly upregulated in the "fibrotic" spots in the D2-*mdx* model. **e** Spatially plotted expression of *Vim*, *Fn1* and *Thbs4* in D2-*mdx* and DBA/2J muscle (**** representing a *p* value < 0.0001). The adjusted p-values were: $1.7e^{-10}$ (*Vim*), $1.8e^{-16}$ (*Fn1*) and $1.5e^{-16}$ (*Thbs4*), respectively. **f** Expression levels of selected genes *Bgn*, *Ctsk* and *Spp1*, which were significantly upregulated in the "calcified" spots compared to the "non-calcified" spots in the D2-*mdx* model (**** representing a *p* value < 0.0001). The exact adjusted p-values were: $1.4e^{-38}$ (*Bgn*), $1.8e^{-29}$ (*Ctsk*) and $5.8e^{-50}$ (*Spp1*), respectively. **g** Expression of the selected genes *Bgn*, *Ctsk* and *Spp1* spatially plotted in the D2-*mdx* and its genetic background matching DBA/2J wildtype. Note: expression values should not be compared across models.

*p* value = 1.44e-38, Fig. 6f, g). *Bgn*, encodes for the ECM proteoglycan biglycan and is therefore also expressed in the connective tissue of healthy muscle sample (Fig. 6g). *Bgn* expression mapped to connective tissue in DBA/2J muscle but not in the D2-*mdx* sample (Fig. 6g), hinting towards changes to the ECM. *Ctsk* was more specifically expressed in calcified areas ($\log_2$(FC) = 1.23, adjusted *p* value = 1.84e-29, Fig. 6f, g) with little-to-no expression in the healthy DBA/2J sample (Fig. 6g-DBA/2J). *Ctsk* encodes for cathepsin K and is used as a marker for osteoclasts and known to be involved in osteogenesis[45,46]. Finally, *Spp1* was highly upregulated in the most severely affected region of the D2-*mdx* muscle section ($\log_2$(FC) = 2.66, adjusted *p* value = 5.78e-50, Fig. 6f, g). *Spp1* is a marker for osteoblasts and osteoclasts and can be secreted as a cytokine in inflammatory processes[47]. It is therefore not only considered as marker of tissue calcification which explains the high expression throughout the D2-*mdx* affected areas.

### Targeted validation of identified markers using smFISH

To validate a subset of the previously identified genes involved in histopathological changes in DMD mouse models, a sensitive, targeted approach with single molecule resolution was used by applying HiPlex smFISH on a consecutive section of the Visium samples. The full panel consisted of: [1] cell type markers (*Cd68*, *Pdgfra*, *Cd34*), markers for [2] regeneration (*Hspg2*, *Myl4*), [3] fibrosis (*Fn1*, *Thbs4*) and [4] calcification (*Bgn*, *Ctsk*) (Supplementary Figs. 10–13). smFISH experiments were performed on a full section; representative areas matching the indicated areas of the Visium HE-stained image were chosen for visualization purposes (Fig. 7a). Fitting the expected profile, based on the differential gene expression analysis, neonatal myosin *Myl4* expression was not detected in the wildtype models and showed highest expression levels in regions of regenerating fibers in the *mdx* mouse model (Fig. 7b). Furthermore, *Thbs4* expression in the wildtype models was restricted to the connective tissue sheet, which follows the Visium data gene expression pattern. In the D2-*mdx* sample, there was increased expression of *Thbs4* in fibrotic areas surrounding the affected muscle fibers (Fig. 7c). Finally, calcification marker *Ctsk* was not expressed in the wildtype mice or milder affected *mdx* mouse model, whereas it showed high expression, specifically in calcified fibers in the D2-*mdx* sample (Fig. 7d). Co-expression of the identified marker genes underlying the histopathological changes in DMD mouse models with cell type specific markers is visualized in Supplementary Figs. 12 and 13. Moreover, biological validation of these findings was done by applying HiPlex smFISH on two independent samples. These experiments confirmed the gene expression patterns previously described from technical validation experiments in consecutive sections of the Visium samples (Supplementary Fig. 15).

### RNA velocity reveals patterns of differentiation

To assess whether the histopathological changes observed in dystrophic muscle are associated with cellular transcriptional reprogramming, we applied RNA velocity on the D2-*mdx* spatial dataset, which displayed the highest histological variation compared to the wildtype counterpart (Fig. 8a).

RNA velocity is defined as the time derivative of gene expression in single cells and has been developed to resolve the potential future state of a cell. RNA velocity recovers information about the direction of a single cell in the transcriptional space by distinguishing between newly transcribed pre-mRNAs (unspliced) and mature mRNAs (spliced), which can be detected in sequencing data[31]. For this analysis we considered each spot on the Visium slide as a single cell. The proportions of reads mapping to spliced and unspliced transcripts (17–23%) fell into the expected range of bulk and single cell RNA sequencing data (also 17–23%[31])[48]. The proportion of unspliced transcripts was lower in the DBA/2J mouse compared to the D2-*mdx* mouse and compared to reference data and is therefore not considered to be representative (Supplementary Fig. 16a, b). The velocities, indicating the direction and strength of change in the transcriptional state of each spot, are displayed on the tissue slide as arrows (Fig. 8c). We found the strongest pattern in spots surrounding the area of inflamed and/or calcified fibers with arrows pointing towards the calcified area (Fig. 8c×d). This implies that cells underlying these spots are transitioning, likely developing a transcriptionally similar profile to those inflamed and/or calcified fibers.

We then sought to identify the genes contributing the most to the estimated RNA velocities. Based on the velocity strength in spots from the inflamed and/or calcified fibers cluster (324 spots), we identified the top genes contributing to the transcriptional profile in each of these spot (see Methods for details) (Fig. 8e). Next, we checked the cell-type specific expression of those highly contributing genes using the reference dataset of McKellar and colleagues (Fig. 8f)[49]. While the analysis showed that myonuclei and other cell types can participate to the directional pattern by transcribing genes such as *Tpm1* and *Vim*, most transitions were based on the presence of other cell types such as macrophages and monocytes expressing *Lgals3*, and FAPs expressing *Fn1*, *Cxcl14* and *Ctsk*.

### Discussion

Dystrophic muscles are exposed to chronic muscle damage, which eventually leads to replacement of muscle fibers by fibrotic and adipose tissues[11]. The presence of cell-types and gene expression signatures have been shown to play key-roles in DMD pathogenesis. So far, it has been impossible to directly link the histological alterations to transcriptional changes. Here we used spatial transcriptomics to overcome this limitation. We identified genes and cell types associated with dystrophic features such as muscle regeneration, fibrosis and calcification, clarifying how FAPs, macrophages and myonuclei participate in the transition of muscle mass into fibro-adipogenic tissue.

Based on the direct link between histology and gene expression in the Visium data, we validated some known biomarkers of muscle regeneration and revealed previously unknown associations. *Myl4* is a known marker for regeneration and it has been previously suggested to be a useful marker for monitoring dystrophic changes[50]. *Sparc* is a less known regeneration marker, but upregulation of the protein product SPARC has been shown to reflect the severity of lesions in DMD and Becker muscular dystrophy biopsies[36,51]. Extensive upregulation of *Sparc* may have an adverse effect on muscle regeneration capabilities[35]. Upregulation of extracellular matrix genes (including *Sparc*) in DMD muscle has also been associated with the dystrophic changes in the muscle such as necrosis, inflammation and

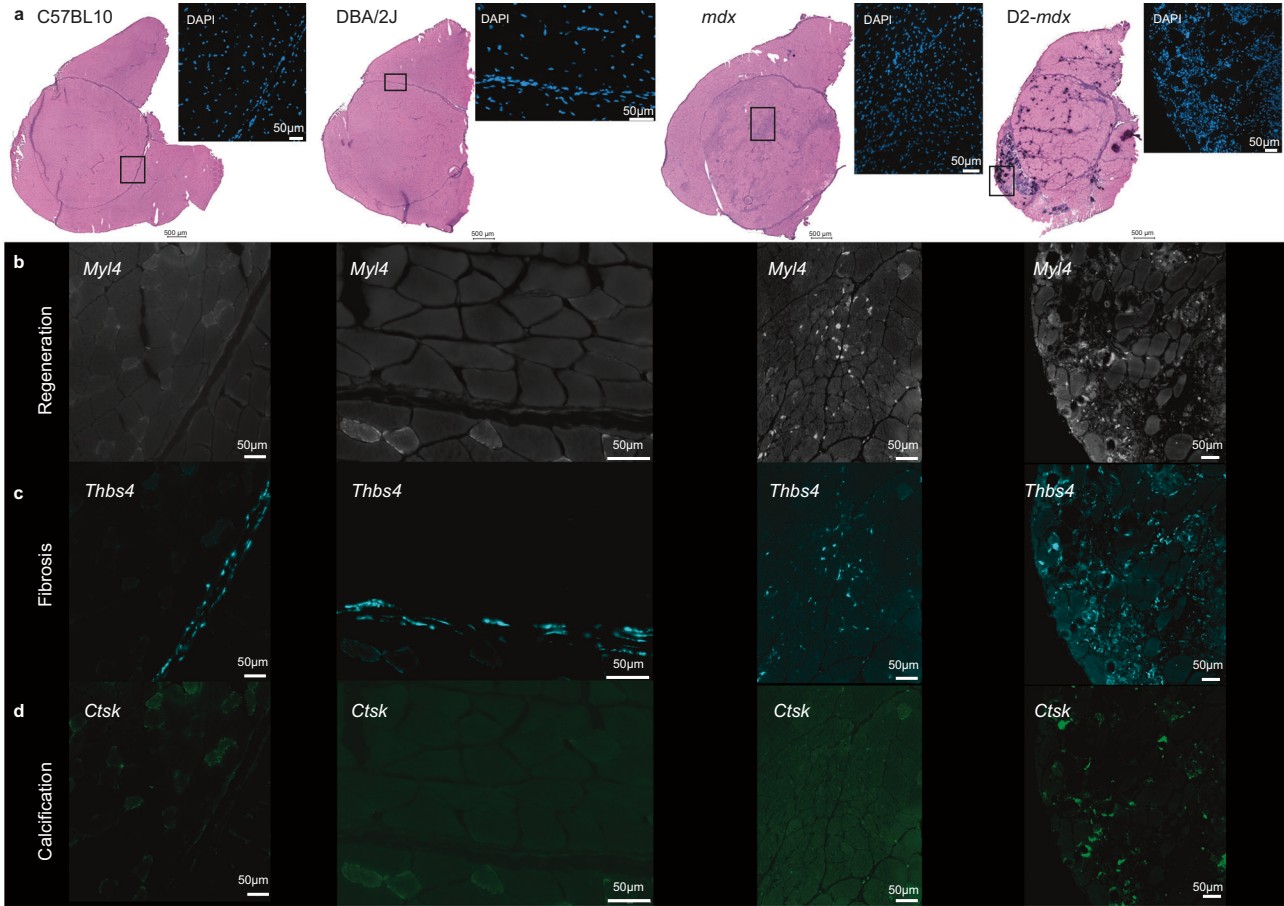

**Fig. 7 | Validation of marker genes using smFISH (RNAscope). a** Visualized areas of smFISH experiment and matching locations in the HE-stained Visium sections. **b** Expression of *Myl4*, a marker of regeneration. **c** Expression of *Thbs4*, a marker of fibrosis. **d** Expression of *Ctsk*, a marker of calcification.

regeneration[50]. Although *Sparc* was identified as a regeneration marker in the *mdx* mouse, we observed high levels of *Sparc* expression in the wildtype (C57BL10) sample. *Sparc* is thought to have contradicting roles depending on its localization. Intracellularly, it modulates the cytoskeletal structure and function in skeletal muscle[52]. Extracellularly, the calcium-binding glycoprotein SPARC, also known as osteonectin, encoded by the *Sparc* gene regulates cell interaction with the extracellular matrix[50,53]. Because of this extracellular activity, high expression levels of *Sparc* in healthy skeletal muscle, especially in the connective tissue sheet, are less surprising. Finally, we identified *Hspg2* as a biomarker for muscle regeneration. The *Hspg2* gene encodes perlecan, a heparan sulfate proteoglycan suggested to encompass several functions in cell growth, tissue organization and differentiation[54]. Perlecan has not been linked to muscular dystrophy before, but our data suggest that it could potentially be used to study regeneration in dystrophic muscle. What is known however, is that a deficiency of perlecan attenuates skeletal muscle atrophy[55]. Indeed, upregulation of *Hspg2* was observed in the necrosis cluster where atrophic muscle fibers were present. Moreover, *Hspg2* has been shown to be responsible for myotonia[56]. However, others suggest that *Hspg2* relates to regeneration even beyond muscle[57]. Further studies are needed to understand the role of *Hspg2* in DMD pathology, but an upregulation of both *Hspg2* and *Myl4* in regions of regeneration in the *mdx* mouse was confirmed by additional experiments in this study.

In the severely affected D2-*mdx* mouse, pathological changes in the dystrophic muscle show partial co-localization, which makes it challenging to identify biomarkers specifically related to one of these histopathological changes. Spatial transcriptomics allowed us to select areas/spots where specific pathological changes are exclusively

present, enabling the identification of genes specific for either fibrosis or calcification.

*Fn1* and *Vim* are well-established fibroblast markers and this study underlines this by re-identifying them as important biomarkers of fibrosis in the D2-*mdx* mouse model. This study was able to confirm these findings on a single molecule level by showing co-expression of these markers on a single tissue section. *Fn1* has been used as a target for therapeutic interventions aiming to reduce fibrotic infiltration in DMD pathology[26,58]. The selected fibrotic markers seem to play important roles in ECM construction and remodeling, which is in line with previous literature where staining of *Vim* has been used and described as a marker for profound ECM changes in dystrophic muscles[59]. Expression of *Thbs4* was predominantly linked to the perimysium in wildtype mice. *Thbs4* encodes thrombospondin-4 (TSP-4), which is thought to function as a structural protein in tendons and connective tissue[60–62]. However, *Thbs4* was found to be upregulated in fibrotic regions and had great overlap with FAP marker *Pdgfra*. In a previous study, TSP-4 overexpression in a dystrophic Drosophila model, rescued the phenotype[63]. This may relate to the tole of TSP-4 in stabilizing the sarcolemma in skeletal myofibers and regulating the production and composition of the ECM[63]. The upregulation of *Thbs4* in the assigned fibrotic spots in the D2-*mdx* sample could either be explained by a compensatory mechanism aiming to stabilize muscle fibers, or to a fibrotic / ECM remodeling phenotype.

*Bgn* is a marker for fibrosis previously described in *mdx* mice and DMD patients[64,65]. More specifically, injection of recombinant human biglycan (rhBGN) has been proposed as a potential therapeutic intervention for DMD following promising studies in *mdx* mice[66]. In other diseases, upregulation of *Bgn* has been linked to contribute to a

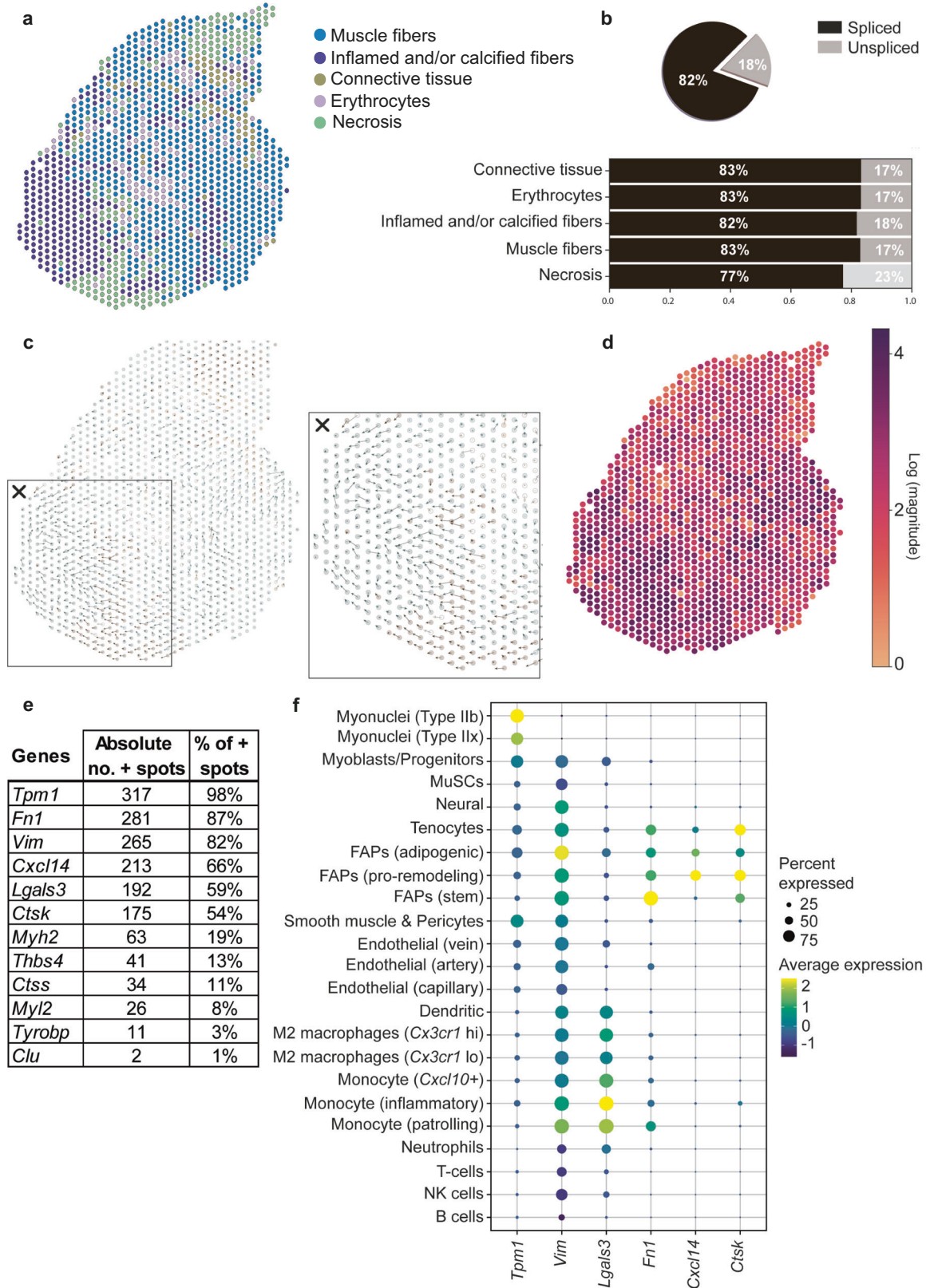

**Fig. 8 | RNA velocity applied on D2-*mdx* muscle shows differentiation patterns in severely affected muscle tissue which is driven by known cell types such as FAPs and macrophages. a** Annotated clusters of D2-*mdx* as described before (**b**) Proportion of spliced/unspliced counts in the D2-*mdx* sample per annotated cluster. **c** Spatial spot-level RNA velocity vectors showing clear differentiation at the severely affected spatial region (box with cross). **d** Magnitude of the RNA velocity at each spot. **e** Summary of the top contributing genes (absolute count and percentage of positive spots) to the differentiation patterns of the severely affected spots (Inflamed and/or calcified fibers). **f** Expression of these top genes in cell types from a reference dataset[49].

pro-osteogenic effect in vascular calcification[67,68]. Moreover, *Bgn*-deficient mice show reduced growth, decreased bone mass and osteoporosis[69]. We think that the role of *Bgn* might not be limited to the fibrotic process but rather to contribute to the calcification/ossification observed in D2-*mdx* mice. On the other hand, the spatial distribution of *Ctsk* expression, which has been used as efficacy outcome after therapeutic intervention aiming to decrease dystrophic calcification[46], has shown to be very specifically expressed in these calcified myofibers in the D2-*mdx* sample, suggesting that *Ctsk* could be a more specific marker of calcification/ossification than *Bgn*. Finally, osteopontin, the product of the *Spp1* gene, has been linked to DMD pathology in different ways. It has been shown that a SNP at the promoter site is associated with disease severity and it has been shown to affect regeneration and inflammation levels[47,70,71]. Another study linked *Spp1* upregulation to vascular smooth muscle cell calcification, supporting the association of this gene in the calcified areas of the D2-*mdx* mouse[72].

While the current spatial resolution of the Visium platform is limited to the 55 µm spot size, we could estimate the cell type contribution using a deconvolution strategy based on existing snRNAseq data. The deconvolution results indicated that multiple cell types (mainly myonuclei) are contributing to the expression profile of the Visium spots, with clear enrichments of specific cell types matching the annotated clusters. The reference snRNAseq dataset, however, was not derived from the same muscle group nor mouse model, representing a limitation of our approach. Despite this limitation we could validate the enrichment of the deconvoluted cell types in specific tissue areas using smFISH for a subset of the cell types (macrophages, FAPs and regenerating myonuclei). The validation step, including biological validation in independent samples, therefore supports the use of the deconvolution approach despite the differences in mouse model, muscle group and age present in the spatial and snRNAseq datasets.

Our proof-of-principle study (including *N* = 1 per mouse model) allowed us to find and link gene expression profiles to clusters that reflected the histology of skeletal muscle samples. Inclusion of one section per animal did not allow for direct comparisons between mouse models. Although we have more than one observation per animal (e.g. multiple spots located in areas of fibrosis in the D2-*mdx* model), the sequencing data cannot be compared directly across models since we cannot estimate if the variation in expression is due to technical differences in tissue and library preparation. To validate the associations found in the discovery we included technical and biological validation using smFISH. However, this study shows that the spatial transcriptomics approach is feasible for muscular dystrophies and how it adds spatial information compared to single-cell approaches, and high-throughput compared to targeted gene specific approaches. The spatial approach we took shows great promise for the identification of biomarkers underlying histopathological changes observed in dystrophic muscle. We have opted for an *N* = 1 approach in our discovery analysis followed by technical validation using an independent technology with higher sensitivity and increased spatial resolution on an adjacent tissue section as well as biological validation in independent mice. Identification of these biomarkers deepens our knowledge on the DMD pathology occurring in these mouse models. In addition to our findings, future studies including multiple individuals per group will allow for direct comparison across dystrophic models.

## Methods

### Animals
Mice were handled according to the guidelines established by the Animal Experiment Committee (Dierexperimenten commissie) of the Leiden University Medical Center (protocol PE.17.246.026). At 10 weeks of age, male C57BL/10ScSn-*Dmd*[mdx]/J (*mdx*), its matched wildtype background C57BL/10ScSnJ (C57BL10), the more severely affected mouse model *D2*.B10-*Dmd*[mdx]/J (D2-*mdx*) and its matched healthy background DBA/2J (DBA/2J) were euthanized by cervical dislocation. A total of six mice were included in the protocol of which one mouse per mouse model was used for Visium analysis and the additional mice were used for biological validation. Mice were euthanized by cervical dislocation. The quadriceps were isolated, embedded in O.C.T. compound (Sakura Finetek USA, Torrance, CA, USA), mounted on a piece of cork and fresh frozen in liquid nitrogen cooled isopentane. Tissue samples were transferred on liquid nitrogen and stored at −80 °C until further processing.

### Visium spatial gene expression library construction
One fresh frozen isolated quadriceps (right-sided) from all four mouse models (C57BL10, DBA/2J, *mdx* and D2-*mdx*) was cryo-sectioned at 10 µm thickness at −21 °C using a CryoStar NX70 cryostat (Thermo Scientific, Waltham, MA, USA). For each mouse model, one cross-section was placed on one of the four 6.5 mm-squared capture areas of a pre-cooled Visium Spatial Gene Expression slide (product code: 1000187, PN: 2000233, 10x Genomics) and adhered to the slide by warming the backside of the slide. The Visium Spatial Gene Expression slide was processed according to the manufacturer's protocols. The slide was transferred on dry-ice and briefly warmed to 37 °C for 1 minute after which fixation in ice-cold methanol was completed for 30 min at −20 °C. Hereafter, the slide was covered with isopropanol at RT for 1 minute and consecutively a hematoxylin and eosin (HE) staining (Agilent Technologies, Santa Clara, CA, USA; Sigma-Aldrich, Burlington, MA, USA) was performed according to the manufacturer's protocol. Images covering the entire capture area were taken with a BZ-X700 microscope (Keyence, Osaka, Japan) using a 10× objective, stitched using the BZ-X700 analyzer software (Keyence) and exported as TIFF files. Permeabilization of the tissue sections was followed after imaging for 15 min. Upon release of the poly-adenylated mRNA from the tissue section, this was captured by barcoded primers on the Visium slide. Through reverse transcription in the presence of template-switching oligo, the captured mRNA was converted to spatially barcoded, full-length cDNA. Following second strand synthesis, a denaturation step released the cDNA from each capture area upon which PCR amplification, with a total of 16 cycles (based on Cq values) was performed. Subsequently, enzymatic fragmentation and size selection (SPRI beads) were used to optimize the cDNA amplicon size. Finally, to generate a sequencing ready indexed library, P5, P7, i7 and i5 sample indexes, and TruSeq Read 2 were added via end repair, A-tailing, adaptor ligation and PCR amplification. The spatial gene expression libraries were sequenced on an Illumina NovaSeq6000 with a target of ~125 million Paired-End reads per sample, or 50,000 reads per spot for each sample.

### Visium data processing and analysis
After sequencing, Illumina's raw data was demultiplexed to fastq files using Space Ranger's pipeline 'mkfastq' (10x Genomics Space Ranger v1.1.0). We used manual alignment of the tissue section and the fiducial frame (microscopic image) using the Loupe Browser (10x Genomics Loupe Browser v4.1.0), followed by generating the spatial feature count matrices using Space Ranger's pipeline 'count'. Further analysis was done in R (version 4.1.1), using the Seurat package (version 4.0.5). We analyzed each tissue section (i.e. mouse model) separately. Mitochondrial genes were filtered out before samples were further processed to overcome bias coming from these genes. The following genes were excluded: *mt-Nd1, mt-Nd2, mt-Co1, mt-Co2, mt-Atp8, mt-Atp6, mt-Co3, mt-Nd3, mt-Nd4l, mt-Nd4, mt-Nd5, mt-Nd6, mt-Cytb*. Furthermore, spots with very few counts or extremely high counts were excluded by filtering each sample separately based on the number of UMIs (nCount) and the number of genes (nFeature). The cut-offs were determined by visual inspection of the violin plots for each sample: for

C57BL10 nCount >= 150 and <= 40,000, nFeature >=150 and <=5000, for DBA/2J nCount >= 100 and <= 20,000, nFeature >=200 and <=4000, for *mdx* nCount >= 200 and <= 40,000, nFeature >=250 and <=5000 and for D2-*mdx* nCount >= 100 and <= 20,000, nFeature >=150 and <=5000. Upon filtering, the spatial datasets were normalized using SCTransform. Hereafter, Principal Component Analysis was applied to the normalized data (RunPCA), and the top 30 principal components were used to generate a neighborhood graph using 20 neighbors (FindNeighbors). Next, data was clustered (FindClusters) with different resolutions (C57BL10 = 0.4, DBA/2J = 0.8, *mdx* = 0.8, D2-*mdx* = 0.4) and UMAP[73] embedding was generated (RunUMAP). The FindAllMarkers (min.pct = 0.1, only.pos = TRUE, min.cells.feature=5) function was used to identify cluster-specific marker genes (Supplementary data file 1). The top 20 markers per cluster genes, based on Bonferroni-corrected p-values, were exported to EnrichR. Here, we looked at enrichment in cell types in the PanglaoDB Augmented 2021 reference dataset to guide the annotation of the clusters. The enriched cell types per cluster were used in the annotation of the clusters together with observation of the histological image underlying these cluster spots. Moreover, the top 50 genes (highest expression) in the cluster 'muscle fibers' for each mouse model was identified and the counts and percentage of spots belonging to the muscle fibers cluster was calculated. This provides us a list of genes accompanied by a percentage of how many spots from that cluster were expressing that gene (summarized in Supplementary data file 2).

## Spot deconvolution using a single nucleus reference dataset

We used SPOTlight[74] to deconvolute each capture location (spot) of the 10X Visium data into cell types. SPOTlight uses seeded non-negative matrix factorization regression to integrate single-nuclei RNA sequencing (snRNAseq) and spatial transcriptomic datasets[75]. SPOTlight learns topic signatures from a reference snRNAseq dataset and uses this to find the optimal weighted combinations of cell types to deconvolute the data from Visium spots into underlying cell types. We used the dataset of Chemello et al., 2020 as a reference snRNAseq dataset as this was the best matched single-nucleus dataset available[33]. This snRNAseq dataset was obtained from the tibialis anterior (TA) of healthy wildtype (C57/BL6N) as well as an affected DMD mouse model (ΔEx51; deletion of *Dmd* exon 51). A total of $n = 70$ cells per cell type and $n = 3000$ highly variable genes were used as input for the deconvolution. Upon completion of the cell type deconvolution using SPOTlight, we calculated the average percentage of each cell type present in the previously annotated clusters. Moreover, we also calculated the average contribution of each cell type to the whole sample per mouse model.

## Visualization and validation of the NMJ population

A BTX staining, co-stained for laminin and DAPI, was performed to visualize NMJs in skeletal muscle tissue sections. The glass slides were defrosted for 30 min at RT followed by a fixation of 2% PFA in PBS for 10 min. Hereafter, the tissue was permeabilized with 0.5% triton PBS for 10 min, washed three times with PBS for 5 min. 2% BSA in PBS was added as a blockbuffer onto the tissue for 1 hour followed by three wash steps with PBS for 5 minutes. Consecutively, we added 1:100 Rabbit-anti-laminin (Abcam, ab11575, Lot GR233309-2) in blockbuffer for 2 hours. Again, followed by three washes with PBS for 5 minutes. Hereafter, the secondary antibody mix (Abcam, goat-anti-rabbit 750, ab175735; Lot GR148346−6, 1:500 dilution) and BTX-488 (donkey-anti-human-IgG 488 with BTX, Invitrogen, Thermo Fisher Scientific, OR, USA, B13422, Lot 2194175, 1:500 dilution) was added for 1.5 hours. Finally, the slides were again washed for three times with PBS for 5 min after which the slides were mounted with Prolong Gold antifade reagent with DAPI (Invitrogen, Thermo Fischer Scientific, Eugene, OR, USA, P36935, Lot 2465351) and covered with a coverslip. Whole-tissue sections were imaged using a ZEISS Axio Scan.Z1 with a 20x/0.8 M27

Plan-Apochromat objective. Detailed imaging settings can be found in Supplementary Table 1.

## Identification and validation of biomarkers using DGE analysis

To identify biomarkers underlying histopathological changes in DMD mouse models, we performed differential gene expression analysis on a selection of spots. The histopathological markers were identified in each model separately. For each histopathological change of interest different marker genes and cut-offs were used to select these spots.

Regeneration was evaluated in the highly regenerative *mdx* mouse. Spots were considered to fall in the category 'regeneration' when they came from one of the two clusters: "regenerated fibers" or "regenerating fibers and inflamed patches" and were expressing either *Myog* and *Igfbp7* or *Myh3* and *Igfbp7* in the count data >0. The so called, non-regenerating category spots, came from the cluster "muscle fibers without CN" and did not express any of these markers. Fibrosis was evaluated in the more severely affected D2-*mdx* mouse. Spots were considered to fall in the 'fibrosis' category when they came from one of the following clusters: "inflamed and/or calcified fibers", "muscle fibers" or "connective tissue" and were expressing either solely *Cd34*, or *Cd34* and *Lox* in the count data >0. Additionally, spots from the "connective tissue" cluster were included in the ´fibrosis´ category when they expressed *Lox* and *Col1a1* to a higher extent with a threshold on the counts >3. The non-fibrotic fiber category was selected based on the "muscle fibers" cluster and no expression of *Cd34* or *Lox* and an expression <2 of *Col1a1*. Finally, calcification was also further investigated in the D2-*mdx* mouse. Spots were included in the 'calcified' category when they fell into the cluster "inflamed and/or calcified fibers" and had an expression >0 of the marker *Mgp*. The noncalcified category came from the cluster "muscle fibers" and had no expression of *Mgp*.

After selection of the categories, we assessed differentially expressed genes between the affected and unaffected spots using the FindMarkers function in Seurat. We used the default parameters in FindMarkers which uses a Wilcoxon Rank Sum test, only testing genes expressed in 10% of the cells in either group, and only testing genes with a $\log_2$(fold-change) > 0.25. Finally, we looked for positive (upregulated) markers in the affected category compared to the non-affected category. The genes were sorted based on fold-change and the significant marker genes were further evaluated (Supplementary data file 3).

The expression pattern of a selection of marker genes was validated using smFISH, HiPlex RNAscope (Advanced Cell Diagnostics Inc., 3960 Point Eden Way Hayward, CA 94545). For this, 10 μm thick consecutive tissue sections of the Visium samples, as well as 10 μm thick tissue sections from independent samples for biological validation, were used and handled according to the manufacturer's instructions for fresh-frozen samples (document number: 324100-UM). Whole-tissue sections were stained for DAPI, *Pdgfra*, *Cd34*, *Cd68*, *Myl4*, *Hspg2*, *Fn1*, *Thbs4*, *Bgn* and *Ctsk* and imaged using a ZEISS Axio Scan.Z1 with a 20x/0.8 M27 Plan-Apochromat objective. Product information on the used probes can be found in Supplementary Table 2 and detailed imaging settings that were used in all three imaging rounds can be found in Supplementary Table 3.

## RNA velocity analysis

In order to apply the RNA velocity[31] analysis, the unspliced and spliced gene expression are required for each spot. We used kallisto/ BUStools[76,77] to map the sequencing reads (fastq) to the reference genome (mm10) while quantifying intronic (unspliced) and exonic (spliced) reads. Next, we applied the RNA velocity pipeline[78,79] (dynamic model implemented in the scvelo python package) to estimate the RNA velocity vector (i.e differentiation) of each spot in its spatial context. For the top 2000 variable genes, high dimensional RNA velocity vectors were calculated using 30 principal component

and 30 neighbors. Next, these vectors were projected on the spatial coordinates of the tissue using the velocity_embedding and the velocity_embedding_stream functions.

To identify the genes mostly underlying the estimated RNA velocities, we considered the high dimensional RNA velocity vectors calculated using the top 2000 variable genes. For each spot, we transformed its high-dimensional RNA velocity vector into a unit vector (magnitude = 1) and we considered the squared value of each gene to represent the contribution of the gene to the RNA velocity of the spot of interest. For the 'Inflamed and/or calcified fibers' of the D2-*mdx* mice (324 spots), we calculated the top 5 contributing genes in each spot, and counted how many times each gene was observed in the top 5 list across all 324 spots.

### Statistics and reproducibility

Samples in this study were selected based on RNA quality (RIN score). No statistical method was used to predetermine sample size. The Visium experiments had a total of $n = 1$ per mouse model and thus no randomization was possible. To validate the findings presented in this study, smFISH experiments were performed on consecutive sections of the Visium included samples. Additionally, two biological replicated were included in the smFISH experiments. All data obtained was used in the data analysis. To find upregulated differentially expressed genes, a one-sided Wilcoxon Rank Sum test was used, Bonferroni-corrected. The adjusted p-values are reported in corresponding figure legends. The investigators were not blinded during the experiments and analysis.

### Reporting summary

Further information on research design is available in the Nature Portfolio Reporting Summary linked to this article.

## Data availability

Raw data files are available in Gene Expression Omnibus (GEO) under accession number "GSE199659". Processed data files are made available on Zenodo under https://doi.org/10.5281/zenodo.7401196. The PanglaoDB dataset (doi:10.1093/database/baz046) was used for gene enrichment analysis through EnrichR in guidance towards cluster annotation. All other data supporting the findings of this study are available within the article and its supplementary files. Any additional requests for information can be directed to, and will be fulfilled by, the lead contact.

## Code availability

The codes used for the analysis in this manuscript are available on GitHub: https://github.com/lauraheezen/SpatialAnalysis_DMD_mice.git. This code can be cited with the following DOI identifier: https://doi.org/10.5281/zenodo.8073645.

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

## Acknowledgements

This project was funded by the Leiden University Medical Center (LUMC).

## Author contributions

L.H. carried out the experiments, analysis and drafted the first version of the manuscript. T.A. contributed by applying RNA velocity on the spatial datasets and writing the methods on this analysis. P.S., A.M., M.v.P., and A.A.R. have jointly supervised the work. All authors provided feedback and comments.

## Competing interests

The authors declare no competing interests.
