## [Peer Review File · Nature Communications]

REVIEWER COMMENTS

Reviewer #1 (Remarks to the Author):

This study did a nice job on the molecular profiling of DMD muscle with spatial transcriptomics. They have revealed several novel genes which were not known to be involved in DMD. The database generated from this work could be informative for the diagnostic strategy of DMD patients in the future.

Here are some suggestions for this manuscript before considering publication.

1) My major concern here is about the resolution of Visium spatial analysis. Each detection spot is about 55um in diameter based on the information from official website. This means the detection spot is about a similar size to the cross section of a mature muscle fiber from adult mice. mRNA profiling from each detection spot in healthy muscle (C57BL10, DBA/2J) may target 1-3 cells. However, in muscles with damage (mdx, D2-mdx), each spot may profile RNA from a larger, and relatively more diverse cell populations due to various fiber size, inflammatory infiltration, fibrosis etc. This may also explain the lower median UMI counts per spot in mdx and D2-mdx. This shouldn't be a big issue for diagnosis purposes as authors have mentioned in the manuscript. However, higher resolution analysis for gene confirmation is highly recommended here once applied to cell type determination. It is necessary to determine whether those candidate genes from "regenerated fiber", "regenerating fiber" and "inflamed/calcified fiber" are truly from muscle fibers, but not due to the RNA pooling from other cell types surrounded. Authors may consider performing FISH for detection of a potential fiber marker, or real-time PCR on purified cell populations for other markers of mononuclear cells.

2) It's also interesting that the NMJ population was only detected in DBA/2J based on Fig1. All four muscle samples should have NMJs, even though they may only be partially functional in mdx and D2-mdx. Can the author validate this detection and make a feasible explanation?

Minor issue:

1) Fig1 labeling does not match with their descriptions in line95.

2) Typo in line 74, line 165.

3) Authors may consider changing Visium spot color in Fig1 to make each one more visible.

4) Hspg2 was previously shown to be involved in neuromyotonia by causing endplate AchE deficiency and NMJ disorganization. Authors may consider citing - Stum, Morgane, et al. "Evidence of a dosage

effect and a physiological endplate acetylcholinesterase deficiency in the first mouse models mimicking Schwartz–Jampel syndrome neuromyotonia." *Human molecular genetics* 17.20 (2008): 3166-3179.

Reviewer #2 (Remarks to the Author):

In the present work a previously described method (Stahl PL, et al., 2016) was applied to characterize spacial transcriptomics in four quadriceps sections that originated from two different mouse models of Duchenne muscular dystrophy (DMD) and their healthy counterparts. Between 1400 and 2400 spots per section were characterized, with median genes per spot varying from 800 to 1800. These primary data were subject to rather complicated statistical analysis, allowing to identify clusters of genes that could be linked to some histological features in the respective sections. The authors suggest that this analysis "could enable to design new diagnostic and therapeutic strategies for DMD".

There are several points that have to be clarified before starting the design of new diagnostic and therapeutic strategies for DMD based on these data.

1. There is only one section per model; these sections are probably made at different heights of the quadriceps (vastus intermedius presents only in one section, C57BL10). Clustering could depend from the position of the slice, not only on the model.
2. Cluster "muscle fibers" is different in C57BL10 and DBA/2J models. In the text of the manuscript, "top markers" for healthy muscle fibers are as following: Myh4, Tpm2 and Ckm (lines 104-105). But in "Supplementary Information" (Tables S1 and S2) top markers are Myh4, Ckm, Tnnt3, Tpm1 for C57BL10 and Myh4, Tmp1, Tnnt3 for DBA/2J. Does it mean that gene expression signatures are different for the healthy muscles in the modes?
3. Surprisingly, cluster "Neuromuscular junction" was detected only in DBA/2J. Are muscles so different in BL10 and DBA/2J mice? This fact is not discussed in the manuscript.
4. Cluster "Connective tissue" was not found in mdx model. The top markers for this cluster are different in C57BL10, DBA/2J and D2-mdx: (Col1a1, Fmod, Col1a2, Thbs4); (Fmod, Col1a1, Col1a2, Sparc); and (Col1a2, Col1a1, Fmod, Chad) respectively. Absence of this cluster in mdx model has to be discussed, as well as differential gene expression in this cluster between other samples.
5. Identification of regeneration biomarkers: one of the criteria to identify genes associated with muscle regeneration was the presence of Myog, Igfbp7 and Myh3 in areas of regeneration and absence of these genes in non-regenerating spots (lines 213-215). Surprisingly, Myh3 is not in the final list of "Differentially expressed gene regeneration" (Table S5).
6. Deconvolution: SnRNAseq dataset used on this step of the analysis is coming from data set obtained from tibialis anterior (not quadriceps) and from a mouse model different from those studied in the present work (Δ Ex51; deletion of Dmd exon 51). These data were obtained on a 4-week old mouse, while in the present work mice of 10 weeks of age were analyzed. The way muscles regenerate varies

between 4 and 10 weeks, which could lead to the misinterpretation of muscle cellular composition obtained as a result of the deconvolution based on ΔEx51 model.

7. RNA velocity reveals patterns of differentiation in severely affected skeletal muscle (1). Some sentences of this paragraph were difficult to understand for me, like : “The resulting velocities are displayed on the tissue slide as arrows where indicating the direction and strength of change in transcriptional the state of each spot” (lines 328-330). What is the meaning of the sentence: “the strongest pattern in spots surrounding the area with inflamed and/or calcified fibers with arrows pointing towards the calcified area” (lines 331-332)? It is difficult to understand how difference in the expression pattern of the neighbors’ spots proves “that cells underlying these spots are transitioning likely developing a transcriptionally similar profile to those inflamed and/or calcified fibers (lines 333-334). Some independent experiments have to be done to confirm this statement.

8. RNA velocity reveals patterns of differentiation in severely affected skeletal muscle (2). The authors found that in the most affected model D2-mdx, cluster “necrosis” was enriched in unspliced transcripts (23% versus 17% in other clusters). They say that these values fell into the expected range of bulk and single cell RNA sequencing data. At the same time they found that the proportion of unspliced transcripts was lower in the DBA/2J mouse compared to the D2-mdx mouse (7% of unspliced transcripts, Figure S6). Is 7% also fell “into the expected range of bulk and single cell RNA sequencing data”? Does this difference (7% versus 20%) corresponds to the difference between normal and dystrophic muscle? If this is the case, this can be reproduced for C57BL10 and mdx mice.

Rebuttal: “Spatial transcriptomics reveal markers of histopathological changes in Duchenne muscular dystrophy mouse models” Heezen, L.G.M.

Reviewer 1 (R1): The only remaining issue is that it is not clear if they performed HiPlex validation on the same sample they used for Visium-seq, which is not ideal. It would be better to use an independent sample. The authors should clarify if the same sample was used and, if so, if it is possible to do their validation experiment on different samples. If not, the authors should write a paragraph about limitations in the discussion, and this could include the possibility that changes may depend on section depth and explain why they used the same sample for validation.

Reviewer 2 (R2): The authors provided exhaustive answers to the questions and conducted important additional experiments. Nevertheless, absence of biological and technical replicates makes difficult to find out the biological significance of the study. Without establishing of these variabilities, the significance of the established gene expression patterns could be compromised. For example, out of 50 genes selected to compare a “muscle fiber cluster” between four models, only 22 are shared. What is the value of 28 that do not overlap, especially for the cluster in healthy strains of mice? Authors' statement: "Low n is often seen in experimental studies like this, for both the spatial and single-cell approaches, as each experiment involves thousands of measurements per sample, allowing for inter-cell comparisons and spot clustering to be simulated" is difficult to accept. Similar studies on different models have already been published (see for example #93 from this manuscript and DOI: <https://doi.org/10.1038/s41588-021-00972-2>). Otherwise, each new model analyzed by the same method should be considered a "proof of principle".

We thank the reviewers for their input and consideration. As both reviewers raised the same point, we will address this at once. In the previous round of review, reviewers suggested to include validation experiments to assess the validity of the identified associations. We have addressed this point in the manuscript by adding several validation experiments in subsequent tissue sections obtained in the same muscles used for the discovery phase. This first layer of technical validation confirmed indeed that the association identified in the discovery phase are not technology dependent. However, those experiment did not address the point on whether the findings are sample specific. In this new version we have therefore included a second layer of biological validation. We performed HiPlex smFISH experiments for the full gene panel used for technical validation on two independent samples of DBA/2J and D2-*mdx* mice. Samples were chosen in order to match age, sex and muscle group to the Visium samples. The results of these experiments show results comparable to the technical validation and they further confirm that the identified genes are associated with the histological alteration.

In the latest version of the manuscript, we have added an additional figure as Supplementary Figure S14 to show the results of the biological validation. Moreover, we added/edited text in the results, discussion and methods sections including details of the biological validation experiments. The added/edited text can be found below:

Results:

“Moreover, biological validation of these findings was done by applying HiPlex smFISH on two independent samples. These experiments confirmed the gene expression patterns previously

described from technical validation experiments in consecutive sections of the Visium samples (Figure S14).”

Discussion:

“The validation step, including biological validation in independent samples, therefore supports the use of the deconvolution approach despite the differences in mouse model, muscle group and age present in the spatial and snRNAseq datasets.”

Methods:

“A total of six mice were included in the protocol of which one mouse per mouse model was used for Visium analysis and the additional mice were used for biological validation.”

and

“For this, 10µm thick consecutive tissue sections of the Visium samples, as well as 10µm thick tissue sections from independent samples for biological validation, were used and handled according to the manufacturer’s instructions for fresh-frozen samples (document number: 324100-UM).”

Reviewers' comments:

Reviewer #1 (Remarks to the Author):

The added validation is reasonable and strengthens the manuscripts.

Reviewer #2 (Remarks to the Author):

My criticism for the present manuscript is due to the use of a very complex method without correctly establishing its limits. Concretely, one slice per model was used for the “Visium analysis”. And these two words include multi-step process, where variations are possible at each step (some of them: 1. From quadriceps isolation to freezing: variation in preparation time can lead to variations in the mRNA integrity; 2. From cryo-sectioning to positioning on the Visium slide: same mRNA integrity problem 3. Visium slide processing: see “Methods” section: a very complicated procedure). All of these steps can introduce variations into the final sets of “gene expression spatial libraries”. These variations are clearly visible in Table 1, where two healthy models have very different average readings per spot (46,436 vs. 86,680) and median genes per spot (1,050 vs. 1,772). One cannot be sure that the differences found between these two models in gene clustering are due to the biological differences or rather to the technical variations. To be sure of the results, it is necessary to analyze the same model at least twice (it is better to have three repetitions of the experiment). Then, if the technical variations allow it, we can speak of biological differences between models. The same applies to the differences between two mouse models with Duchenne muscular dystrophy and between healthy mice and mouse models with muscular dystrophy. Probably, one can rely on some of the data for large regions that are histologically different, but that's pretty trivial.

What is missing in the current revised version: technical replicates of “Visium analysis”

Response: “Spatial transcriptomics reveal markers of histopathological changes in Duchenne muscular dystrophy mouse models” Heezen, L.G.M.

We thank the reviewer for their comment on, and consideration of, our manuscript (NCOMMS-22-10715B entitled *Spatial transcriptomics reveal markers of histopathological changes in Duchenne muscular dystrophy mouse models*, by Heezen et al). Please find our response to the comment raised by the reviewer below.

Reviewer 2 (R2): My criticism for the present manuscript is due to the use of a very complex method without correctly establishing its limits. Concretely, one slice per model was used for the “Visium analysis”. And these two words include multi-step process, where variations are possible at each step (some of them: 1. From quadriceps isolation to freezing: variation in preparation time can lead to variations in the mRNA integrity; 2. From cryo-sectioning to positioning on the Visium slide: same mRNA integrity problem 3. Visium slide processing: see “Methods” section: a very complicated procedure). All of these steps can introduce variations into the final sets of “gene expression spatial libraries”. These variations are clearly visible in Table 1, where two healthy models have very different average readings per spot (46,436 vs. 86,680) and median genes per spot (1,050 vs. 1,772). One cannot be sure that the differences found between these two models in gene clustering are due to the biological differences or rather to the technical variations. To be sure of the results, it is necessary to analyze the same model at least twice (it is better to have three repetitions of the experiment). Then, if the technical variations allow it, we can speak of biological differences between models. The same applies to the differences between two mouse models with Duchenne muscular dystrophy and between healthy mice and mouse models with muscular dystrophy. Probably, one can rely on some of the data for large regions that are histologically different, but that’s pretty trivial.

What is missing in the current revised version: technical replicates of “Visium analysis”

Response: The reviewer raises the point that the discovery phase performed on N=1 with Visium slides is not sufficient as more animals are needed to compare across models. The reviewer clarifies in multiple instances how the comparison between mice is not acceptable due to the limited number of samples per group.

While we agree in principle that a higher number of mice could be beneficial for the discovery phase, this argument does not hold for the spatial transcriptomics analysis in our set up which we want to clarify step-by-step:

1. The reviewer argues that we cannot compare between models due to technical variation. We fully agree with this statement as sample-to-sample variations could affect the results. Also, no statistical analysis is valid with $N = 1$. For these reasons, we did not compare between mice. Throughout the manuscript, we only compared areas of interest (e.g., fibrosis vs healthy muscle) within a single section for each mouse separately. As such, all analyses are performed per mouse model independently (differential expression, deconvolution and RNA velocity) and the results are not affected by read depth and other sample-to-sample variation. We acknowledge that this could be better explained or highlighted in the text and we have adapted the text of the manuscript accordingly.
2. In spatial analysis, due to high costs and challenges in tissue preparations, $N = 1$ is acceptable for discovery provided that independent validation of the findings is performed. This is what we did, based on the suggestions of both reviewers, by including validation experiments using fluorescent in-situ hybridization (FISH), first in the adjacent section of the same mice (technical validation) and in the last version of the manuscript also in independent individuals (biological validation). We would

further like to point out that *Nature Communications* (as well as other journals) has recently published papers where the N = 1 approach was also adopted in the discovery phase, using the same exact Visium technology that we applied in our study¹⁻³.

To address the reviewer's concerns and to make sure the limitations of our work are clearly communicated, we have now included additional sentences in a limitations paragraph in the discussion section to further clarify these points and to outline what is acceptable (compare within a sample) and what is not (compare between samples) in the N=1 setting. We further stress the importance of complementing the study with robust technical and biological validation as we provide in our study.

Changes to the Manuscript:

Added the following to the Methods section:

- Further analysis was done in R (version 4.1.1), using the Seurat package (version 4.0.5). *We analyzed each tissue section (i.e. mouse model) separately.*
- To identify biomarkers underlying histopathological changes in DMD mouse models, we performed differential gene expression analysis on a selection of spots. *The histopathological markers were identified in each model separately.*

Added the following in the Results:

- The caption of figure 5: (c) *Expression levels of selected genes Myl4, Sparc and Hspg2 in the two categories with an enrichment in the "regenerating" spots in the mdx model (**** representing a p-value <0.0001).* (d) *Spatially plotted expression of Myl4, Sparc and Hspg2 in mdx and C57BL10 muscle. Note: expression values should not be compared across models.*
- The caption of figure 6: (d) *Violin plot showing expression levels of selected genes Vim, Fn1 and Thbs4 which were significantly upregulated in the "fibrotic" spots in the D2-mdx model.* (e) *Spatially plotted expression of Vim, Fn1 and Thbs4 in D2-mdx and DBA/2J muscle (**** representing a p-value <0.0001).* (f) *Expression levels of selected genes Bgn, Ctsk and Spp1 which were significantly upregulated in the "calcified" spots compared to the "non-calcified" spots in the D2-mdx model (**** representing a p-value <0.0001).* (g) *Expression of the selected genes Bgn, Ctsk and Spp1 spatially plotted in the D2-mdx and its genetic background matching DBA/2J wildtype. Note: expression values should not be compared across models.*

Added the following sentences (in red) to the Discussion:

Our proof-of-principle study (including N=1 per mouse model) allowed us to find and link gene expression profiles to clusters that reflected the histology of skeletal muscle samples. *Inclusion of one section per animal did not allow for direct comparisons between mouse models. Although we have more than one observation per animal (e.g. multiple spots located in areas of fibrosis in the D2-mdx model), the sequencing data cannot be compared directly across models since we cannot estimate if the variation in expression is due to technical differences in tissue and library preparation. To validate the associations found in the discovery we included technical and biological validation using smFISH.* However, this study shows that the spatial transcriptomics approach is feasible for muscular dystrophies and how it adds spatial information compared to single-cell approaches, and high-throughput compared to targeted gene specific approaches. The spatial approach we took shows great promise for the identification of biomarkers underlying histopathological changes observed in dystrophic muscle. *We have opted for an N=1 approach in our discovery analysis followed by technical validation using an independent technology with higher sensitivity and increased spatial resolution on an adjacent tissue section as well as biological validation in independent mice.* Identification of these biomarkers deepens our knowledge on the DMD pathology occurring in these mouse models. *In addition to our findings, future studies including multiple individuals per group will allow for direct comparison across dystrophic models.*

Cites References:

1. Parigi, S. M. *et al.* The spatial transcriptomic landscape of the healing mouse intestine following damage. *Nat. Commun.* 2022 131 **13**, 1–16 (2022).
2. Mantri, M. *et al.* Spatiotemporal single-cell RNA sequencing of developing chicken hearts identifies interplay between cellular differentiation and morphogenesis. *Nat. Commun.* 2021 121 **12**, 1–13 (2021).
3. Hunter, M. V., Moncada, R., Weiss, J. M., Yanai, I. & White, R. M. Spatially resolved transcriptomics reveals the architecture of the tumor-microenvironment interface. *Nat. Commun.* 2021 121 **12**, 1–16 (2021).

REVIEWER COMMENTS

Reviewer #2 (Remarks to the Author):

The authors performed a great work in improving the writing of the manuscript. Nevertheless, there are some points which are not clear.

1. It is not clear why “the results are not affected by read depth and other sample-to-sample variation” if “analyses are performed per mouse model independently”. Beside other reasons, results can be affected by border effects on slides, or precision in density of capture oligonucleotides on the slide.

2. The high cost does not mean N=1 and cannot be an argument. Some questions concerning the independent Validation by using smFISH, HiPlex RNAscope. For example, to validate regeneration markers, the authors used Myl4 (which was already known) and Hspg2 (but not Sparc, found by Visium experiment). While in the Visium experiment Hspg2 was detected in C57BL10, its expression was not confirmed by smFISH. Citation: “Sparc and Hspg2 expression was not negligible in wildtype mice. Especially Sparc expression was observed at high levels across the tissue, but primarily in the connective tissue sheet and VI muscle” (Small note to Figure 5: mdx and C57BL10 slides on this picture are not on the same scale). What is the value of these regeneration markers (Hspg2 and Sparc) if they are found in non-regenerating samples? And what conclusion can be drawn from this analysis: the found markers work only for mdx mice? Only for this slide of quadriceps of mdx mouse? Or there is a difference between non-regenerating muscle fibers in mdx mice and healthy mice?

Data from “Supplementary File 4 (List of differentially expressed genes) was used to identify (confirm?) many clusters, but it is not clear how it was obtained. For example, you find (line 230) “Differential gene expression analysis between these two groups of spots revealed markers of regeneration (top 50 markers summarized in supplementary file 4). This file contains 429 genes ranged by their adjusted P-values, and three selected by the authors genes (Myl4, Sparc and Hspg2) are on the positions 376, 81, and 170 respectively. The same table was used to distinguish calcified and non-calcified spots and to identify fibrotic markers, but the positions of the selected gene in the table is not explained.

Small remarks:

1. The statement that “we did not compare between mice” is not fully true, because authors (and readers as well) tend to compare similar clusters between mice. For example (from line 267): “Finally, Thbs4 seemed more specific for fibrosis rather than being present in all damaged areas, given the more restricted expression pattern in D2-mdx ($\log_2(\text{FC}) = 1.32$, adjusted p-value = 1.46×10^{-16}), whereas localization of Thbs4 expression is limited to perimysial tissue in the healthy DBA/2J sample”

2. From line 251: “To identify fibrotic markers, spots expressing known fibrotic genes such as Cd34, Lox and/or Col1a1 were compared to spots belonging to the muscle fibers cluster where fibrotic genes were not expressed”. CD34 is not a fibrotic marker.

3. The following clusters in the manuscript are mentioned: healthy muscle fibers: connective tissue; slow twitch muscle fibers; erythrocytes; immune response; neuromuscular junction. Dystrophic samples: regenerated fibers; regenerating fibers with inflamed patches; necrotic muscle fibers with infiltrating macrophages; adipocytes; prominent inflammation and calcification; necrotic fibers, but the complete information about genes per cluster is given only for 4 clusters (Supplementary File 2): (1) Erythrocytes, (2) Connective tissue, (3) VI muscle fibers, (4) Immune response.

Response: “Spatial transcriptomics reveal markers of histopathological changes in Duchenne muscular dystrophy mouse models” Heezen, L.G.M.

We thank the reviewer for their consideration of, and comments on, our manuscript (NCOMMS-22-10715B entitled *Spatial transcriptomics reveal markers of histopathological changes in Duchenne muscular dystrophy mouse models*, by Heezen et al). Please find our response to the comments raised by the reviewer below in blue.

Reviewer 2 (R2): The authors performed a great work in improving the writing of the manuscript. Nevertheless, there are some points which are not clear.

R2 comment 1: It is not clear why “the results are not affected by read depth and other sample-to-sample variation” if “analyses are performed per mouse model independently”. Beside other reasons, results can be affected by border effects on slides, or precision in density of capture oligonucleotides on the slide.

We thank the reviewer for the question. The results (i.e. genes associated with histological observations) are not affected by read depth across samples because we do not compare gene expression across samples. All comparison are performed within each sample separately. As we describe in our methods: We analyzed each tissue section (i.e. individual mouse) separately. Therefore, differences in read-depth across samples do not influence the results.

Read depth does also not affect the comparisons within each sample because data are normalized. Here, we followed the widely used standard pipeline including: filtering low quality spots and normalization using the SCT method implemented in the Seurat package. The normalization step ensures that technical variation is removed from the data (within each tissue section) and regresses out technical variation due to UMI counts and percentage of mitochondrial reads. To identify marker genes, the comparison of normalized gene expression is then performed among several spots (e.g. 192 regenerating spots vs 105 non-regenerating spots) within each section separately. This analysis results in clear gene expression markers from normalized gene expression data per sample. Such marker genes are therefore not affected by differences in read-depth across and within samples.

In this comment the reviewer also raised the point that “results can be affected by border effects on slides, or precision in density of capture oligonucleotides on the slide”. This is not reported by the manufacturing company nor by the multiple research groups that have extensively been using the Visium gene expression slides. We have no hints in our data that make us reason these might be possible problems and it is outside of the scope of this paper to benchmark the oligonucleotide printing fidelity during the manufacturing process.

R2 comment 2a: The high cost does not mean N=1 and cannot be an argument. Some questions concerning the independent Validation by using smFISH, HiPlex RNAscope. For example, to validate regeneration markers, the authors used Myl4 (which was already known) and Hspg2 (but not Sparc, found by Visium experiment). While in the Visium experiment Hspg2 was detected in C57BL10, its expression was not confirmed by smFISH. Citation: “Sparc and Hspg2 expression was not negligible in wildtype mice. Especially Sparc expression was observed at high levels across the tissue, but primarily in the connective tissue sheet and VI muscle” (Small note to Figure 5: mdx and C57BL10 slides on this picture are not on the same scale). What is the value of these regeneration markers (Hspg2 and Sparc) if they are found in non-regenerating samples? And what conclusion can be drawn from this analysis: the found markers work only for mdx mice? Only for this slide of quadriceps of mdx mouse? Or there is a difference between non-regenerating muscle fibers in mdx mice and healthy mice?

Thank you for picking up the scale difference in Figure 5, this has now been amended.

We see the point raised by the reviewer about the regeneration markers. *Hspg2* was chosen as it was largely enriched in the areas of regeneration compared to areas where regenerating fibers were not observed in *mdx* muscle. *Hspg2* encodes the perlecan protein, which has multiple binding partners in the extracellular matrix as it interacts with laminin, collagen IV and other extracellular proteins. Staining of perlecan is not regeneration specific in healthy mice. Sarcolemma and endomysial areas are stained in undamaged areas with antibodies targeting perlecan. Staining of perlecan has been however shown to increase a few days after damage in correspondence to regenerating areas supporting an upregulation of *Hspg2* expression after damage, which normalizes after the regeneration is complete¹. Gene expression changes anticipate protein synthesis, therefore supporting the association of *Hspg2* with regenerating areas that we identified in the Visium discovery phase and validated by smFISH. The fact that *Hspg2* is expressed in healthy mice is expected as perlecan is needed in healthy muscles and lack of perlecan in muscle has been shown to affect muscle fiber composition and it has been associated with metabolic alterations². Our interpretation is therefore that *Hspg2* is expected to be widely expressed in healthy skeletal muscle and locally upregulated during muscle regeneration. This observation is confirmed by the distribution of spots expressing *Myl4* and *Hspg2* (see table R1 below). Table R1 shows how *Myl4* and *Hspg2* are expressed in WT mice and how *Hspg2* expression is more widespread compared to *Myl4*. The percentage of spots co-expressing both markers is quite low with only 2.65% of spots expressing both marker genes in WT muscle. In *mdx* mice however, the percentage of spots expressing both marker genes is 16.49%. The relationship between regeneration and co-expression of *Myl4* and *Hspg2* is also evident in figure R1, where the area of regeneration in *mdx* mice is highly enriched in spots expressing both genes. Co-localization of *Myl4* and *Hspg2* was also confirmed by smFISH as shown in Supplementary Figure S12 as indicated by the triangle (◄) and pound (#) symbol on the image.

Figure R1. Co-expression of identified regeneration markers *Myl4* and *Hspg2*. (a) The absolute number of spots that are expressing *Myl4*, *Hspg2* or both as well as the percentage of spots that are expressing *Myl4*, *Hspg2* or both. (b) Spots highlighted that are co-expressing *Myl4* and *Hspg2* in the C57BL10 and *mdx* mouse.

Regarding the validation of *Sparc*, we agree with the reviewer that it would have been nice to also validate the expression of *Sparc* and other genes identified during the discovery phase by using HiPlex smFISH; however, we could only fit a restricted number of genes into the HiPlex validation panel, considering probes availability and the possibility of combining them into a single assay. Indeed, not all probes are available in multiple channels (as these are linked to spectral colors to be imaged by the microscope). The inclusion of *Sparc* in the assay would have forced us to leave out other genes. We

therefore focused on *Myl4* and *Hspg2* to actually confirm a well-described marker for regeneration such as *Myl4* and extend the knowledge of *Hspg2*.

Changes to the manuscript, regarding the co-expression of *Myl4* and *Hspg2*:

- Figure R1 was added as Supplementary Figure S14
- The following text was added to the results section:
 - o "*Hspg2* encodes the perlecan protein, which in healthy muscle is present in undamaged sarcolemmal and endomysial areas. However, perlecan expression was found to be increased a few days after muscle damage in regenerating areas, followed by a normalization upon completion of the regeneration process 39,40. Co-expression of *Hspg2* and *Myl4* was particularly present in areas where active muscle regeneration is taking place in the *mdx* model (Figure S12 and S14)."

R2 comment 2b: Data from "Supplementary File 4 (List of differentially expressed genes) was used to identify (confirm?) many clusters, but it is not clear how it was obtained. For example, you find (line 230) "Differential gene expression analysis between these two groups of spots revealed markers of regeneration (top 50 markers summarized in supplementary file 4). This file contains 429 genes ranged by their adjusted P-values, and three selected by the authors genes (*Myl4*, *Sparc* and *Hspg2*) are on the positions 376, 81, and 170 respectively. The same table was used to distinguish calcified and non-calcified spots and to identify fibrotic markers, but the positions of the selected gene in the table is not explained.

We would like to thank the reviewer for pointing out that the text still mentions that Supplementary File 4 lists the 'top 50 markers', while we now in fact list all differentially expressed marker genes. We have adapted the text accordingly.

Regarding the genes and their position in the list of significant differentially expressed marker genes, we provide one Supplementary File 4, which consists of multiple tabs. These tabs are labelled according to the analysis performed as e.g. 'regeneration', 'fibrosis' and 'calcification' followed by 'DEgenes'. The numbers mentioned by Reviewer 2 for the genes *Myl4*, *Sparc* and *Hspg2* are derived from the 'calcification' list and thus place very low in this specific list as these are not top marker genes for calcification. When consulting the correct list (regeneration tab within Supplementary File 4), one would find the beforementioned genes (*Myl4*, *Sparc* and *Hspg2*) in spot 9, 14 and 15 respectively.

Changes to the manuscript, regarding the top 50 marker genes and cited text by R2:

- The following text was added:
 - o "Differential gene expression analysis between these two groups of spots revealed markers of regeneration (all significant differentially expressed markers are summarized in supplementary file 4) such as *Myl4*, *Sparc* and *Hspg2*, which showed little-to-no expression in the non-regenerating spots (Figure 5c)."

R2 small remarks

Remark 1: The statement that "we did not compare between mice" is not fully true, because authors (and readers as well) tend to compare similar clusters between mice. For example (from line 267): "Finally, *Thbs4* seemed more specific for fibrosis rather than being present in all damaged areas, given the more restricted expression pattern in D2-*mdx* ($\log_2(\text{FC}) = 1.32$, adjusted p-value = $1.46e-16$), whereas localization of *Thbs4* expression is limited to perimysial tissue in the healthy DBA/2J sample".

With this statement, we meant that we did not compare statistically between mice.

Remark 2: From line 251: “To identify fibrotic markers, spots expressing known fibrotic genes such as Cd34, Lox and/or Col1a1 were compared to spots belonging to the muscle fibers cluster where fibrotic genes were not expressed”. CD34 is not a fibrotic marker.

CD34 is a well-established marker for fibroadipogenic progenitor cells (FAPs) as it expressed on the surface of these cell types next to *PDGFR α* and *SCA1*. These FAPs are suggested to be the main effectors for adipogenic and fibrotic infiltration in muscular dystrophies and thus has been chosen as an important gene for selection of fibrotic spots³⁻⁶. CD34 has been further related to fibrotic and scarring events in other diseases⁷.

Remark 3: The following clusters in the manuscript are mentioned: healthy muscle fibers: connective tissue; slow twitch muscle fibers; erythrocytes; immune response; neuromuscular junction. Dystrophic samples: regenerated fibers; regenerating fibers with inflamed patches; necrotic muscle fibers with infiltrating macrophages; adipocytes; prominent inflammation and calcification; necrotic fibers, but the complete information about genes per cluster is given only for 4 clusters (Supplementary File 2): (1) Erythrocytes, (2) Connective tissue, (3) VI muscle fibers, (4) Immune response.

Supplementary File 2 indeed contains the marker genes for all clusters. The Excel file consists of 4 tabs (one per mouse model), each tab annotated by mouse model which contains the clusters present in that specific mouse model. The regenerated fibers cluster is for example present in the mdx tab but it is absent in the C57BL10 tab.

Of note, there is no ‘slow twitch muscle fibers’ cluster annotated in our dataset (see Figure 1). The VI muscle cluster consists of slow twitch muscle fibers, and this is then also reflected in the genes expressed in this cluster (Supplementary File 2, tab ‘C57BL10 clusters’).

References:

1. Casar, J. C., Cabello-Verrugio, C., Olguin, H., Aldunate, R., Inestrosa, N. C., & Brandan, E. (2004). Heparan sulfate proteoglycans are increased during skeletal muscle regeneration: requirement of syndecan-3 for successful fiber formation. *Journal of cell science*, 117(1), 73-84.
2. Yamashita, Y., Nakada, S., Yoshihara, T., Nara, T., Furuya, N., Miida, T., ... & Arikawa-Hirasawa, E. (2018). Perlecan, a heparan sulfate proteoglycan, regulates systemic metabolism with dynamic changes in adipose tissue and skeletal muscle. *Scientific reports*, 8(1), 7766.
3. Molina, T., Fabre, P., & Dumont, N. A. (2021). Fibro-adipogenic progenitors in skeletal muscle homeostasis, regeneration and diseases. *Open biology*, 11(12), 210110.
4. Giordani, L., He, G. J., Negroni, E., Sakai, H., Law, J. Y., Siu, M. M., ... & Le Grand, F. (2019). High-dimensional single-cell cartography reveals novel skeletal muscle-resident cell populations. *Molecular cell*, 74(3), 609-621.
5. Brown, J., Greaves, M. F., & Molgaard, H. V. (1991). The gene encoding the stem cell antigen, CD34, is conserved in mouse and expressed in haemopoietic progenitor cell lines, brain, and embryonic fibroblasts. *International immunology*, 3(2), 175-184.
6. Marinkovic, M., Fuoco, C., Sacco, F., Perpetuini, A. C., Giuliani, G., Micarelli, E., ... & Cesareni, G. (2019). Fibro-adipogenic progenitors of dystrophic mice are insensitive to NOTCH regulation of adipogenesis. *Life Science Alliance*, 2(3).
7. Pu X, Zhu P, Zhou X, He Y, Wu H, Du L, Gong H, Sun X, Chen T, Zhu J, Xu Q, Zhang H. (2022) CD34⁺ cell atlas of main organs implicates its impact on fibrosis. *Cell Mol Life Sci*. 79(11):576.

REVIEWERS' COMMENTS

Reviewer #3 (Remarks to the Author):

The authors have fully addressed all previous comments and remarks, and have made the appropriate changes to the manuscript. Therefore, I support it for publication in its current form.